# UP2You: Fast Reconstruction of Yourself from Unconstrained Photo Collections

**Zeyu Cai**[1,2] **Ziyang Li**[2] **Xiaoben Li**[1] **Boqian Li**[1] **Zeyu Wang**[3] **Zhenyu Zhang**[2†] **Yuliang Xiu**[1†]

[1]Westlake University    [2]Nanjing University
[3]The Hong Kong University of Science and Technology (Guangzhou)
[†]Shared Corresponding Author

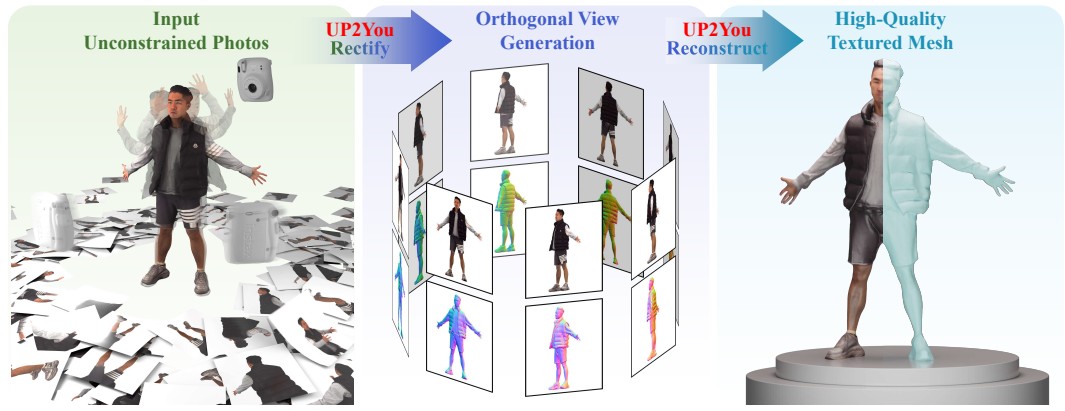

Figure 1: **Overview of UP2You.** Our method reconstructs high-quality, textured 3D clothed portraits from unconstrained photo collections. It robustly handles highly diverse and unstructured inputs by rectifying them into orthogonal multi-view images and corresponding normal maps, making them compatible with traditional reconstruction algorithms.

## ABSTRACT

We present UP2You, the first tuning-free solution for reconstructing high-fidelity 3D clothed portraits from extremely unconstrained in-the-wild 2D photos. Unlike previous approaches that require "clean" inputs (e.g., full-body images with minimal occlusions, or well-calibrated cross-view captures), UP2You directly processes raw, unstructured photographs, which may vary significantly in pose, viewpoint, cropping, and occlusion. Instead of compressing data into tokens for slow online text-to-3D optimization, we introduce a *data rectifier* paradigm that efficiently converts unconstrained inputs into clean, orthogonal multi-view images in a single forward pass within seconds, simplifying the 3D reconstruction. Central to UP2You is a pose-correlated feature aggregation module (PCFA), that selectively fuses information from multiple reference images w.r.t. target poses, enabling better identity preservation and nearly constant memory footprint, with more observations. We also introduce a perceiver-based multi-reference shape predictor, removing the need for pre-captured body templates. Extensive experiments on 4D-Dress, PuzzleIOI, and in-the-wild captures demonstrate that UP2You consistently surpasses previous methods in both geometric accuracy (Chamfer-15%↓, P2S-18%↓ on PuzzleIOI) and texture fidelity (PSNR-21%↑, LPIPS-46%↓ on 4D-Dress). UP2You is efficient (1.5 minutes per person), and versatile (supports arbitrary pose control, and training-free multi-garment 3D virtual try-on), making it practical for real-world scenarios where humans are casually captured. Both models and code will be released to facilitate future research on this underexplored task.

## 1 INTRODUCTION

Reconstructing 3D clothed humans from **unconstrained photo collections**, like the personal albums (Fig. 2-Left), is a challenging and largely unexplored research frontier. Unlike prior tasks such as single-image 3D reconstruction [23, 48, 65, 92, 93], monocular video-based reconstruction [18, 28, 35], or multi-view 3D reconstruction [55, 64, 101], this problem is distinguished by the highly unstructured nature of the input: appearance information is present but scattered across photos

where subjects are often partially captured or occluded, and camera as well as body poses are rarely synchronized. As a result, establishing accurate 2D-to-3D correspondences is extremely difficult, even with the help of most advanced off-the-shelf human-centric estimators (i.e., camera, body pose, landmarks, geometric cues, etc). In contrast, traditional 3D reconstruction algorithms typically assume "clean captures" (i.e., full-body capture with simple poses, synchronized cameras, etc), where well-aligned 2D-to-3D correspondences can be readily established using the estimators above.

Two potential strategies to address above challenges: **1) Data Compressor**: Crop and group photos into local and global patches (e.g., head, full-body) [102], or segment input photos into multiple assets (e.g., garments, hair, face, accessories) [94], then compress these patches or assets into learnable tokens, and finally assemble them as text prompt to generate 3D humans via text-to-3D techniques [62]; **2) Data Rectifier**: Convert the incoming "dirty or incomplete captures" into clean and complete ones, e.g., orthogonal orbit views with canonical poses, which are easier to reconstruct with traditional 3D reconstruction algorithms. Essentially, the data compressor operates mainly at the representation level, without substantially improving the generative model's ability to ensure 3D consistency and identity preservation — a limitation noted in PuzzleAvatar [94] as "unpredictable hallucination." The data rectifier, however, refines not only the input data but also the generative model's prior, via continued training on synthetic multi-view renderings of high-fidelity 3D clothed humans, enabling more consistent 3D reconstruction in terms of both identity and viewpoint, from unconstrained photographs. UP2You falls in the second category, as shown in Fig. 2.

PuzzleAvatar [94] is the representative of the first strategy, it first "decompose" the unconstrained photos into multiple asset soups, all of which are linked with unique learned tokens via DreamBooth [70], then it "compose" these assets into a 3D full-body representation via score-distillation sampling (SDS) [62], where the 3D reconstruction task is reformulated as a text-to-3D task, bypassing explicit canonicalization. However, this process takes hours since both DreamBooth fine-tuning and SDS-based optimization are time-consuming and unstable, see Fig. 2. Additionally, ground-truth SMPL-X meshes are needed for initialization, as predicting shape parameters from unconstrained photo collections is non-trivial. Regarding the second strategy — converting inputs into orthogonal orbit views —— some attempts [23, 48, 60] have been made. However, these methods are restricted to single-image inputs and cannot fully leverage the multiple unconstrained photos. Essentially, these methods act more as "data inpainters" [84] — synthesizing unseen views from seen capture — rather than as "data rectifiers" that unify the messy observations into structured output. Designed mainly for constrained inputs (i.e., a single image with full-body coverage), these methods cannot handle unconstrained photos or scale up the reconstruction accuracy with the number of inputs.

To the best of our knowledge, UP2You is the first work to unlock the "data rectifier" strategy on unconstrained photo collections, directly transforming raw unconstrained photo collections into orthogonal views while faithfully preserving subject identity. This is not a trivial extension of prior arts, as it 1) requires effectively aggregating information from multiple unconstrained inputs, which may vary significantly in terms of body poses, camera viewpoints, croppings, and occlusions; 2) must be efficient enough to process varying numbers of input photos (ranging from one to dozens) without incurring significant computational overhead; and 3) needs to overcome the dependency on ground-truth body shapes, which are often unavailable in real-world scenarios.

Specifically, UP2You aggregates ReferenceNet features [27], extracted from unconstrained photos according to body poses, via the proposed Pose-Correlated Feature Aggregation (PCFA) module. This module implicitly learns correlation weights between unconstrained reference images and target pose conditions (i.e., SMPL-X normal maps). Guided by these correlation maps, PCFA uses an optimized `topk` strategy to selectively aggregate the most informative image features for generating each orthogonal view. As a result, the memory footprint remains nearly constant regardless of the number of input photos, enabling effective and efficient information fusion.

To get rid of the dependence on ground-truth body shapes, we design a shape predictor based on perceiver structure [34, 46] to regress SMPL-X shape parameters directly from unconstrained photo collections. Lastly, with another MV-Adapter [32] to generate multi-view normal maps, followed by mesh carving and texture baking [48], UP2You reconstructs high-quality textured meshes from unconstrained photos in 1.5 minutes. We evaluate our generation results on PuzzleIOI, 4D-Dress, and self-collected in-the-wild datasets. Our method surpasses other state-of-the-art approaches in both geometric accuracy (Chamfer-15%↓, P2S-18%↓ on PuzzleIOI) and texture fidelity (PSNR-21%↑,

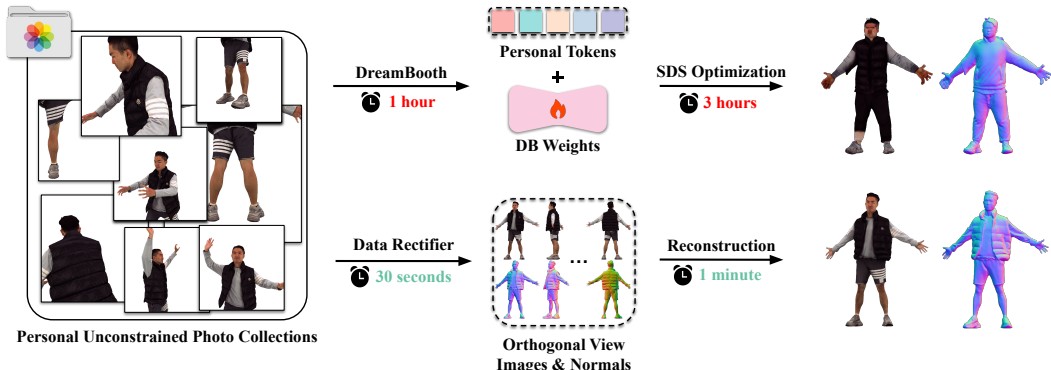

Figure 2: **Paradigm differences between previous works and UP2You. Top:** Previous works like PuzzleAvatar [94] and AvatarBooth [102] compress unconstrained photos into implicit personal tokens and DreamBooth weights [70] through fine-tuning, then generate 3D humans via SDS optimization [70]. **Bottom:** UP2You directly rectifies unconstrained photo collections into orthogonal view images and normals, then reconstructs textured human meshes, achieving superior quality while reducing processing time from 4 hours to 1.5 minutes.

LPIPS-46%↓ on 4D-Dress), while also demonstrating flexibility and superior generalization for single-image reconstruction, and enabling 3D virtual try-on application, all without extra training.

Our main contributions w.r.t. the prior arts are as follows:

- **Efficient**. As Fig. 2 shows, unlike previous DreamBooth + SDS paradigm (>4 hours), UP2You acts as a "data rectifier" instead, to directly generate "clean" multi-views from "dirty" unconstrained inputs in one forward pass (<15 secs). It can process one, several, or dozens of photos with a nearly constant memory footprint. The full pipeline, including multi-view normal generation plus mesh carving and texture baking, completes in 1.5 minute.
- **Effective**. Thanks to the PCFA module, which selectively aggregates the most informative regions from the reference images for synthesizing target views, UP2You significantly outperforms prior SOTAs (PuzzleAvatar, AvatarBooth, PSHuman) in both geometry accuracy and texture fidelity, and delivers consistent shape and identity regardless of input forms or pose conditions. Notably, the reconstruction quality even scales up with more unconstrained inputs, echoing the principle of *The More You See in 2D, the More You Perceive in 3D* [21].
- **Versatile**. PuzzleAvatar requires an A-posed body template with ground-truth shape for 3D initialization, while UP2You is flexible to random pose control, directly regresses body shapes from unconstrained photos, and inherently supports multi-garment 3D virtual try-on, for free.

## 2 RELATED WORK

### 2.1 3D CLOTHED HUMAN RECONSTRUCTION

The field of 3D clothed human reconstruction has been extensively studied over the past few decades. Early methods primarily focused on reconstructing human geometry and texture from dense multi-view image captures [35, 52, 61]. Subsequent research has broadened the scope to include full-shot monocular video inputs [18, 19, 28, 90], enabling more flexible and accessible data acquisition. Recent advances in generative models, particularly diffusion models [25, 42, 69, 81], and the emergence of SDS-based 3D human generators [30, 44, 50, 53, 86, 86], have further propelled the field. An increasing number of video-based human reconstruction approaches now leverage learned generative priors to address common challenges in real-world video captures, such as occlusions [19, 59], view inconsistencies [36], and poor texture details [83].

Such generative priors, learned from large-scale datasets, play a more crucial role for the inherently ill-posed problem of 3D human reconstruction, especially when the input data is sparse or incomplete. The most sparse input format is a single image [23, 31, 33, 48, 65, 67, 71, 72, 92, 93, 107]. In essence, it can be regarded as a "conditional generation" problem [31], since large portions of the geometry — such as the unseen backside and occluded regions — must be plausibly inferred or synthesized from the visible pixels. Building on this "reconstruction as conditional generation" paradigm, numerous works have further advanced the field [3, 15, 48, 105]. Apart from multi-view posed captures, full-shot monocular video, and single image, numerous works have sought to expand the range of input modalities, for example, by incorporating dual front-back captures [38, 55] or multi-view unposed full-body images [29, 66, 97, 101, 108] to improve reconstruction fidelity and completeness.

Despite these advances, existing methods still fall short of handling truly "unconstrained" photos — those with partial views, occlusions, extreme camera viewpoints, dynamic body poses, and inconsistent aspect ratios. Accurately estimating body shape [12, 43, 45, 89, 98, 104] from such unconstrained photo collections is nearly impossible. Moreover, given multiple "dirty" reference images, image-based HMR methods often fail to deliver consistent results. This inconsistency manifests as significant variations in the predicted body shapes for the same subject — some reconstructions may appear unnaturally thin, others excessively fat, and some may completely fail, especially in cases of partial or occluded inputs. As shown in Tab. 2 and Fig. 7, it becomes challenging to determine which, if any, of the predicted shapes truly represent the subject.

In contrast, UP2You addresses these data format constraints by functioning as a comprehensive "data rectifier," directly transforming unstructured or "dirty" inputs into orthogonal "clean" views, with consistent 3D and identity, that can be seamlessly utilized for robust 3D reconstruction.

## 2.2 UNCONSTRAINED PHOTOS TO 3D

Most real-world data is inherently unstructured, presenting significant challenges for 3D reconstruction tasks that require reliable spatial correspondences. The earliest work in "Unconstrained Photos to 3D" can be traced back to Photo Tourism [80], which reconstructs 3D scenes from large collections of Internet photos. Recent advances in neural rendering and generative models have further advanced this field, enabling more robust and realistic 3D reconstructions from unstructured image collections [10, 49, 87]. However, these methods primarily focus on rigid objects or scenes and cannot be directly applied to 3D clothed human reconstruction, which involves highly articulated and non-rigid structures. A critical open question is how to effectively extract and aggregate identity features from unconstrained photos — not only for general objects [41, 103, 109], but especially for dynamic humans — and reproduce them in a 3D-consistent manner. Several works on subject-driven image generation [2, 4, 13, 14, 16, 40, 70, 73, 85], as well as ID-consistent 2D human portrait generation [9, 63, 75, 84, 95], are discussed in the *Sup.Mat.* (Appendix B). However, these methods are primarily designed for 2D image generation and lack the mechanisms to ensure cross-view consistency or the precise latent feature aggregation necessary for high-fidelity 3D reconstruction.

The most relevant works addressing this challenge are PuzzleAvatar [94] and AvatarBooth [102]. Both first employ few-shot personalization [4, 70], as Total Selfie [9] and RealFill [84], to distill identity information from unconstrained photos into a customized diffusion model, as unique tokens. Subsequently, guided by these unique tokens, they utilize Score Distillation Sampling (SDS) [7, 37, 62, 100] to optimize a neural-based 3D representation [56, 76]. In short, the entire pipeline of these methods can be summarized as "unconstrained photos → personalized diffusion models with learned specialized tokens → SDS-based Text-to-3D". However, fine-tuning diffusion models and optimization-based SDS methods are extremely time-consuming. Moreover, these fine-tuning approaches act as a form of lossy compression: the strong priors of diffusion models often override subject-specific features, leading to a loss of identity and fine-grained details, or even introducing unpredictable hallucinations. In contrast, UP2You is a tuning-free method that faithfully reconstructs 3D humans from unconstrained photos in just 1.5 minutes, while well preserving human identities.

## 3 METHOD

Our objective is to reconstruct a high-quality textured mesh from unconstrained photos with unknown camera parameters and human poses. To this end, we first generate orthogonal full-body images from the unconstrained inputs, conditioned on SMPL-X normal maps that contain both camera and pose information (Sec. 3.1). Next, we utilize these orthogonal multi-view RGB images to generate corresponding multi-view normal maps, which serve as geometric cues for detailed mesh reconstruction (Sec. 3.2). To handle in-the-wild images without SMPL-X annotations, we further introduce a body shape estimator capable of inferring human body shape by integrating information from a handful of unconstrained photos (Sec. 3.3).

## 3.1 ORTHOGONAL MULTI-VIEW IMAGES GENERATION

To tackle orthogonal multi-view image generation from unconstrained photo collections, we adopt MV-Adapter [32] as our backbone (introduced in Appendix C). MV-Adapter integrates ReferenceNet

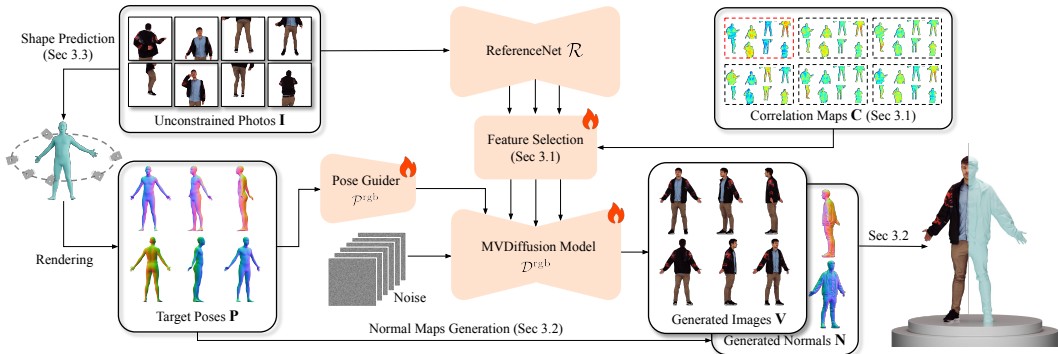

Figure 3: **Pipeline of UP2You.** Given unconstrained input photos $\mathbf{I}$, we first predict the SMPL-X shape parameters (Sec. 3.3) and initialize the SMPL-X mesh with predefined pose and expression parameters. We then generate orthogonal view images $\mathbf{V}$ based on $\mathbf{I}$ and SMPL-X normal rendering $\mathbf{P}$ with the proposed PCFA method—predict correlation maps $\mathbf{C}$ and select most informative features (Sec. 3.1). Finally, we produce multi-view normal maps $\mathbf{N}$ from $\mathbf{P}$ and $\mathbf{V}$, and reconstruct the final textured mesh (Sec. 3.2).

$\mathcal{R}$ [27] as the reference image encoder and incorporates raymaps into the diffusion UNet as view conditions, enabling the synthesis of six orthogonal views. For our task, we use orthogonal SMPL-X normal maps as view conditions. Unlike the original MV-Adapter, which handles only single-image inputs, our approach extends it to process multiple unconstrained photos.

As shown in Fig. 3, given $N$ unconstrained reference images $\mathbf{I} = \{\mathbf{I}_1, ..., \mathbf{I}_N\}$ of a person in the same outfit, our goal is to synthesize $M$ orthogonal target views $\mathbf{V} = \{\mathbf{V}_1, ..., \mathbf{V}_M\}$, each conditioned on a corresponding SMPL-X normal map $\mathbf{P} = \{\mathbf{P}_1, ..., \mathbf{P}_M\}$. To extract the most informative features for each target view, we introduce the Pose-Correlated Feature Aggregation (PCFA) module, which predicts correlation maps $\mathbf{C} = \{\mathbf{C}_1^i, ..., \mathbf{C}_N^i\}_{i=1}^M$ between reference and target views (see Fig. 4). Based on $\mathbf{C}$, PCFA select features for each target viewpoint for the generation of orthogonal views $\mathbf{V}$.

**Correlation Map Prediction.** Using all reference features for ortho-view generation is computationally intensive, as memory usage grows with the number of unconstrained references. However, many reference pixels are irrelevant for a given target view (e.g., back-view references for front-view synthesis). Therefore, we adaptively determine each reference's contribution based on the target pose to reduce computational cost.

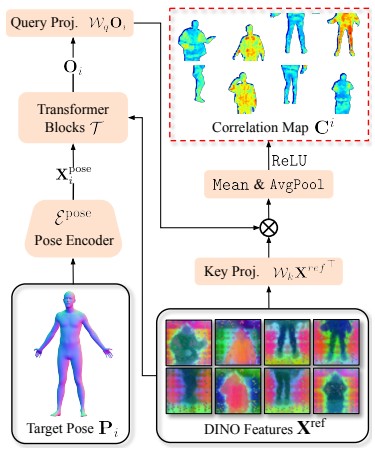

Figure 4: **Pose-Dependent Correlation Map.** Correlation is colored as Higher → Lower.

To achieve this, we disentangle human-specific identity features from viewpoint correlation information in the unconstrained reference inputs. Drawing inspiration from [26, 39], we predict correlation maps for reference images conditioned on target poses, as illustrated in Fig. 4. For each target pose $\mathbf{P}_i, i \in \{1, 2, ..., M\}$, we estimate a correlation map that indicates the pixel-wise relevance of each reference image for generating the corresponding view. Specifically, we employ a pose image encoder $\mathcal{E}^{\text{pose}}$ and a DINOv2 [57] model $\mathcal{E}^{\text{ref}}$ to extract features from the target pose image and all reference images: $\mathbf{X}_i^{\text{pose}} = \mathcal{E}^{\text{pose}}(\mathbf{P}_i)$ and $\mathbf{X}^{\text{ref}} = \mathcal{E}^{\text{ref}}(\mathbf{I})$, where $\mathbf{X}^{\text{ref}}$ represents the concatenation of all DINOv2 outputs $\{\mathbf{X}_j^{\text{ref}}\}_{j=1}^N$. Subsequently, we feed both $\mathbf{X}_i^{\text{pose}}$ and $\mathbf{X}^{\text{ref}}$ into a transformer block $\mathcal{T}$ that comprises layers of self-attention and cross-attention, where $\mathbf{X}_i^{\text{pose}}$ functions as the *query*, *key*, and *value* in self-attention operations, and as the *query* in cross-attention operations, while $\mathbf{X}^{\text{ref}}$ serves as both *key* and *value* in cross-attention operations. Through $\mathcal{T}$, an output feature $\mathbf{O}_i = \mathcal{T}(\mathbf{X}_i^{\text{pose}}, \mathbf{X}^{\text{ref}})$ that integrates reference information relevant to the target pose is produced. We derive the image correlation map $\mathbf{C}^i$ by computing the attention map between $\mathbf{O}_i$ and $\mathbf{X}^{\text{ref}}$:

$$\mathbf{A}^i = \frac{\mathcal{W}_q \mathbf{O}_i \times \mathcal{W}_k \mathbf{X}^{\text{ref}^\top}}{\sqrt{d}}, \tag{1}$$

$$\begin{aligned}
\mathbf{C}^i &= [\mathbf{C}_1^i, \mathbf{C}_2^i, ..., \mathbf{C}_N^i] \\
&= \texttt{ReLU}(\texttt{AvgPool}(\texttt{mean}(\mathbf{A}^i))),
\end{aligned} \tag{2}$$

Here, $\mathcal{W}_q$ and $\mathcal{W}_k$ are learnable projection matrices applied to $\mathbf{O}_i$ and $\mathbf{X}^{\text{ref}}$, respectively. The resulting attention map $\mathbf{A}^i \in \mathbb{R}^{l \times Nhw}$ captures the relevance between the target pose and reference features. To obtain the final reference correlation scores, we compute the mean along the first dimension of $\mathbf{A}^i$ using $\texttt{mean}(\cdot) : \mathbb{R}^{l \times Nhw} \to \mathbb{R}^{Nhw}$. In this context, $l$ is the token number of $\mathbf{O}_i$, $h$ and $w$ denote the height and width of $\mathbf{X}^{\text{ref}}$, and $d$ is the feature dimension of $\mathcal{W}_q\mathbf{O}_i$ and $\mathcal{W}_k\mathbf{X}^{\text{ref}}$. We further apply $\texttt{AvgPool}$ to smooth the predicted correlation map and $\texttt{ReLU}$ to suppress negative values.

The correlation maps of PCFA are based on fine-grained semantic correlation between target bodies and DINO features of references. Unlike previous methods [26, 39] that depend on landmark similarity, our correlation map encodes richer outfit details, enabling more accurate reconstruction.

**Feature Selection.** The predicted correlation maps enable PCFA to selectively aggregate the most informative reference features for each target view. Specifically, we utilize ReferenceNet $\mathcal{R}$ as the reference image encoder to extract multi-scale reference features $\mathbf{F} = \{\mathbf{F}_1, \mathbf{F}_2, ..., \mathbf{F}_L\}$, where $L$ is the number of layers. For each target pose $\mathbf{P}_i$ and the reference feature $\mathbf{F}_k \in \mathbb{R}^{NS_k \times c}$ at layer $k$, we first interpolate the corresponding correlation map $\mathbf{C}^i \in \mathbb{R}^{Nhw}$ to get $\hat{\mathbf{C}}^i = \texttt{Interp}_k(\mathbf{C}^i)$ that aligns with the spatial dimensions of $\mathbf{F}_k$. Here $S_k$ denotes the spatial size of $\mathbf{F}_k$, and $\texttt{Interp}_k(\cdot) : \mathbb{R}^{Nhw} \to \mathbb{R}^{NS_k}$ denotes the interpolation operator.

We then select the most relevant reference features $\hat{\mathbf{F}}_k^i$ for view $\mathbf{P}_i$ based on $\hat{\mathbf{C}}^i$. Specifically, we employ the $\texttt{topk}$ selection strategy to obtain the selected indices of $\mathbf{F}_k$:

$$[k_1^i, k_2^i, ..., k_{\gamma S_k}^i] = \texttt{sort}(\texttt{topk}(\hat{\mathbf{C}}^i)[: \gamma S_k]), \tag{3}$$

where $[k_1^i, k_2^i, ..., k_{\gamma S_k}^i]$ are the indices of the selected features, $\texttt{topk}(\cdot)$ returns the top $\gamma S_k$ indices, and $\gamma$ controls the proportion of features retained. To preserve spatial order, we apply $\texttt{sort}(\cdot)$. Using these indices, we extract the selected reference features $\hat{\mathbf{F}}_k^i \in \mathbb{R}^{\gamma S_k \times c}$:

$$\hat{\mathbf{F}}_k^i = \mathbf{F}_k[k_1^i, k_2^i, ..., k_{\gamma S_k}^i] \cdot \hat{\mathbf{C}}_i[k_1^i, k_2^i, ..., k_{\gamma S_k}^i]. \tag{4}$$

Given the aggregated reference features $\hat{\mathbf{F}} = \{\hat{\mathbf{F}}_k^1, \hat{\mathbf{F}}_k^2, ..., \hat{\mathbf{F}}_k^M\}_{k=1}^L$, we synthesize the orthogonal multi-view images as $\mathbf{V} = \mathcal{D}^{\text{rgb}}(\hat{\mathbf{F}}, \mathcal{P}^{\text{rgb}}(\mathbf{P}))$, where $\mathcal{D}^{\text{rgb}}$ is our multi-view image generation model and $\mathcal{P}^{rgb}(\cdot)$ is the pose guider that encodes the pose condition into $\mathcal{D}^{\text{rgb}}$.

### 3.2 Normal Map Generation and Mesh Reconstruction

For multi-view reconstruction (MVS) [51, 54, 91], we generate multi-view clothed normal maps $\mathbf{N}$ from the generated images $\mathbf{V}$, conditioned on target poses $\mathbf{P}$, and reconstruct the mesh using both $\mathbf{V}$ and $\mathbf{N}$.

**Normal Map Generation.** To ensure multi-view consistency and provide strong geometric cues for normal map generation, we follow [93] and incorporate SMPL-X normal renderings as additional conditions. As Fig. 5 shows, we also adopt MV-Adapter as the backbone of clothed normal generator $\mathcal{D}^{\text{normal}}$. We utilize the generated orthogonal RGB views $\mathbf{V}$ as reference inputs, and employ the pose guider $\mathcal{P}^{\text{normal}}(\cdot)$ to incorporate multi-view pose conditions. The multi-view clothed normal maps are then generated via $\mathbf{N} = \mathcal{D}^{\text{normal}}(\mathbf{V}, \mathcal{P}^{\text{normal}}(\mathbf{P}))$.

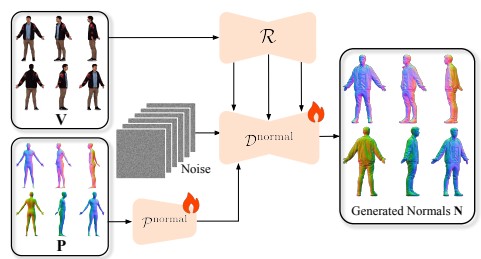

Figure 5: **Normal Map Generation Pipeline.** The main input difference with Fig. 3 is the generated multi-view orthogonal images $\mathbf{V}$, instead of unconstrained inputs $\mathbf{I}$.

| | PuzzleIOI | | | | | | 4D-Dress | | | | | | in-the-wild | |
|---|---|---|---|---|---|---|---|---|---|---|---|---|---|---|
| | PSNR↑ | SSIM↑ | LPIPS↓ | Chamfer↓ | P2S↓ | Normal↓ | PSNR↑ | SSIM↑ | LPIPS↓ | Chamfer↓ | P2S↓ | Normal↓ | CLIP-I↑ | DINO↑ |
| AvatarBooth | 16.879 | 0.860 | 0.1544 | 6.635 | 6.697 | 0.0274 | 18.186 | 0.850 | 0.1718 | 6.846 | 6.978 | 0.0311 | 0.878 | 0.619 |
| PuzzleAvatar | 21.664 | 0.916 | 0.0639 | 3.204 | 3.165 | 0.0150 | 21.376 | 0.887 | 0.1081 | 1.956 | 2.045 | 0.0170 | 0.907 | 0.742 |
| Ours (Image) | 23.896 | 0.926 | 0.0545 | - | - | - | 25.848 | 0.920 | 0.0576 | - | - | - | 0.972 | 0.932 |
| Ours (Mesh) | 24.539 | 0.940 | 0.0474 | 2.724 | 2.605 | 0.0115 | 25.540 | 0.918 | 0.0654 | 1.140 | 1.119 | 0.0122 | 0.971 | 0.916 |

Table 1: **Quantitative Comparison with Baselines.** UP2You achieves the best texture fidelity, geometry accuracy, and perception similarity.

**Mesh Carving and Texture Baking.** Starting from the initial SMPLX mesh, we refine mesh details using the generated $\mathbf{N}$ and project per-vertex colors from $\mathbf{V}$, following PSHuman [48]. To better preserve hand geometry, we replace the hand region with that from the initial mesh as in ECON [93], and then perform texture baking using the generated multi-view RGB images.

### 3.3 MULTI-REFERENCE SHAPE PREDICTOR

The initial SMPL-X mesh is critical to the entire UP2You pipeline, as it provides the pose condition $\mathbf{P}$ for multi-view generation and serves as the basis for mesh reconstruction. SMPL-X mesh $\mathbf{T} \in \mathbb{R}^{10754 \times 3}$ are defined as $\mathbf{T}(\beta, \theta, \psi)$, where $\beta, \theta, \psi$ are shape, pose, and expression parameters respectively. While the target pose and expression of the SMPL-X template can be predefined (e.g., T-pose or A-pose with neutral expression), the body shape parameters must be estimated from unconstrained input images. Existing shape predictors [23, 48, 65] are typically designed for single-image scenarios and struggle to effectively leverage multiple unconstrained references.

To address this limitation, we introduce a multi-reference shape predictor, $\mathcal{S}$, as illustrated in Fig. 6. The prediction process is formulated as $\beta^{\text{pred}} = \mathcal{S}(\tau, \mathbf{X}^{\text{ref}})$, where $\beta^{\text{pred}}$ denotes the predicted shape parameters, $\tau$ are learnable query tokens, and $\mathbf{X}^{\text{ref}}$ are DINOv2 features extracted from the reference images. Our shape predictor $\mathcal{S}$ employs a perceiver-style architecture [34, 46] that can use query tokens to effectively aggregate multi-view information. The prediction head is a lightweight transformer, similar to the camera head design in [87].

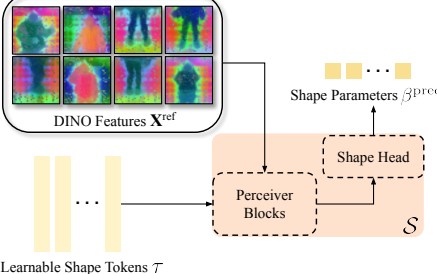

Figure 6: **Multi-reference Shape Predictor.**

Overall, through the shape predictor, multi-view image & normal generator, and mesh carving & texture baking steps, UP2You generates textured 3D humans from unconstrained photo inputs. See the detailed flowchat in *Sup.Mat.*'s Appendix D.5.

## 4 EXPERIMENTS

### 4.1 SETTINGS

**Dataset.** We train our multi-view image generation, normal map generation, and shape prediction models on the THuman2.1 [99], Human4DiT [74], 2K2K [20], and CustomHumans [24] datasets. For evaluation, we use the PuzzleIOI [94] and 4D-Dress [88] datasets as test sets. To further validate our approach, we collect an in-the-wild (in-the-wild) dataset comprising 12 distinct identities. Details on dataset selection and processing procedures are provided in Appendix D.2.

**Baselines.** We comprehensively compare UP2You with 1) album-to-human reconstruction methods, including PuzzleAvatar [94] and AvatarBooth [102]. Since single-view reconstruction is a special case of the unconstrained setting, we also include the leading 2) single-view method, PSHuman [48], in our comparisons. To ensure fair evaluation and isolate the impact of pose estimation errors, we provide ground truth SMPL-X parameters for all baseline methods. 3) For shape prediction, we present the first approach to estimate SMPL-X shape parameters from multiple unconstrained inputs. We compare our shape predictor with two single-input methods: Semantify [17], which is specifically designed for shape prediction, and PromptHMR [89], a state-of-the-art human mesh recovery method. Unless stated otherwise, results on PuzzleIOI and 4D-Dress use 12 reference images. 4) For in-the-wild, we use all available references (8–12) for each identity. Additional model and training details are in *Sup.Mat.*'s Appendices D.1 and D.3.

**Metrics.** For PuzzleIOI and 4D-Dress (with textured 3D GT), we report geometric metrics (Chamfer, P2S, Normal map L2) and image quality metrics (PSNR, SSIM, LPIPS). For in-the-wild, we use perceptual similarity (CLIP-I, DINO) between generated and frontal reference. Shape prediction is assessed by vertex-to-vertex (V2V) distance on all datasets. More details in *Sup.Mat.*'s Appendix D.4.

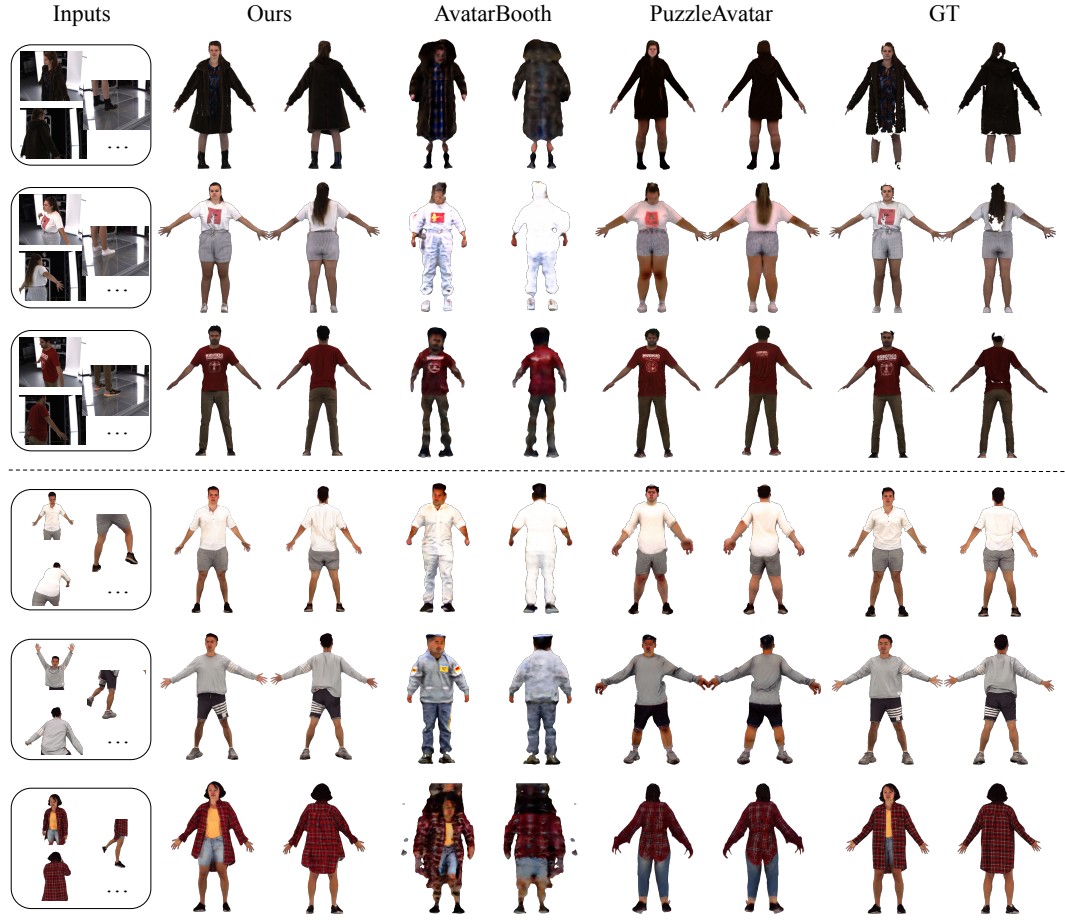

Figure 8: **Qualitative Comparisons on PuzzleIOI and 4D-Dress.** See more 360-degree results in *Sup.Mat.*'s *video*.

## 4.2 COMPARISONS

**Quantitative Results.** The quantitative results in Tab. 1 show that UP2You consistently surpasses all baselines across both 2D and 3D evaluation metrics on the PuzzleIOI and 4D-Dress datasets. Importantly, UP2You also achieves strong perceptual quality scores on the in-the-wild dataset, demonstrating its robustness and effectiveness in real-world unconstrained scenarios.

For single-view reconstruction, Tab. 3 shows that UP2You outperforms PSHuman on all 2D and 3D metrics. This is expected, as single front-view input is a special case of the unconstrained multi-view scenario for which UP2You is designed. Training on the more challenging unconstrained task enables our model to generalize well and excel in the simpler constrained setting.

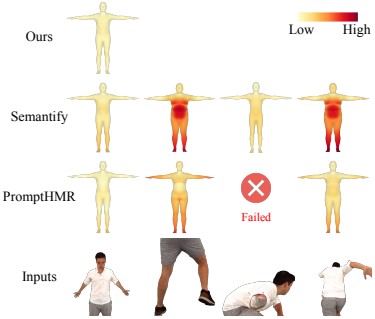

Figure 7: **Shape Prediction Error Map.**

As shown in Tab. 2 and Fig. 7, our shape predictor outperforms single-view methods [17, 89], achieving more accurate and consistent results. Single-input baselines show high variance and instability, especially with partial input or failed detections. Leveraging multiple inputs, our method delivers more robust shape prediction, with performance further improving as more unconstrained references are used. Furthermore, Table 2 also shows that the perceiver transformer architecture is better than simple MLPs for the shape predictor.

| Num of | Semantify | | PromptHMR | | MLP | Ours |
|---|---|---|---|---|---|---|
| References | Mean↓ | Var↓ | Mean↓ | Var↓ | | |
| 3 | 11.087 | 4.234 | 9.212 | 10.370 | 8.819 | **7.967** |
| 6 | 11.066 | 5.706 | 9.661 | 17.465 | 8.046 | **7.427** |
| 9 | 10.978 | 6.424 | 9.403 | 18.218 | 8.275 | **7.403** |
| 12 | 11.097 | 6.597 | 9.287 | 19.418 | 8.336 | **7.399** |

Table 2: **V2V(↓) Comparions of Shape Prediction Reuslts.**

| | PSNR↑ | SSIM↑ | LPIPS↓ | Chamfer↓ | P2S↓ | Normal↓ |
|---|---|---|---|---|---|---|
| PSHuman | 24.134 | 0.905 | 0.0895 | 2.759 | 2.926 | 0.0189 |
| Ours Mesh | **26.651** | **0.935** | **0.0527** | **0.927** | **0.949** | **0.0096** |

Table 3: **Comparison of Single-Image based Reconstruction.**

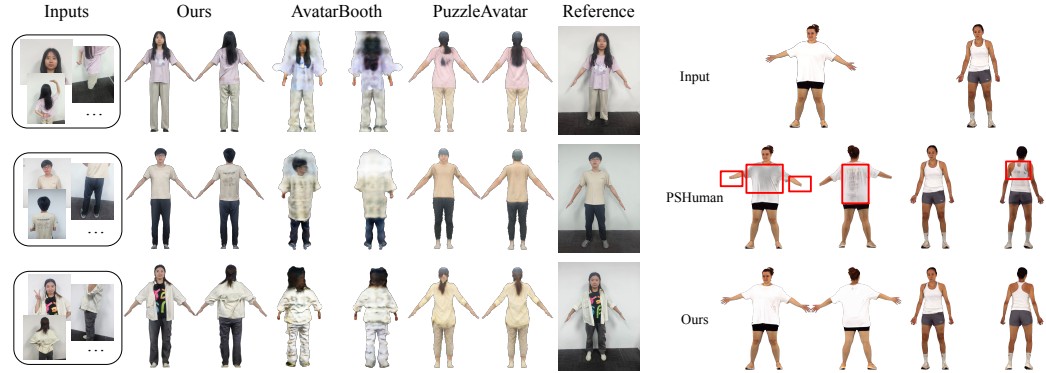

Figure 9: **Qualitative Comparisons on in-the-wild Data.**

Figure 10: **UP2You vs. PSHuman.**

**Qualitative Results.** The qualitative comparisons in Fig. 8 and Fig. 9 show that UP2You achieves high-fidelity, reference-faithful 3D reconstructions with strong realism and detail preservation. In contrast, baselines like AvatarBooth and PuzzleAvatar often fail to capture fine facial details and produce blurrier, less realistic results with poor subject-specific consistency. Figure 10 shows single-view 3D human reconstruction comparisons. Our method generalizes well to single-view inputs, producing visually comparable results to PSHuman, but with more accurate limb reconstruction due to consistent multi-view guidance. More visual comparisons and results are in Appendices E.1 and G.

| | Feature Aggregation | | | | | Image Encoder | | | PuzzleIOI | | | 4D-Dress | | |
|---|---|---|---|---|---|---|---|---|---|---|---|---|---|---|
| | Mean | Concat | Corr. | sum | topk | CLIP | DINOv2 | Ref Net | PSNR↑ | SSIM↑ | LPIPS↓ | PSNR↑ | SSIM↑ | LPIPS↓ |
| Ours | ✗ | ✗ | ✓ | ✗ | ✓ | ✗ | ✗ | ✓ | **23.896** | **0.926** | **0.0545** | **25.848** | **0.920** | **0.0576** |
| A. | ✓ | ✗ | ✗ | ✗ | ✗ | ✗ | ✗ | ✓ | 17.412 | 0.864 | 0.1227 | 19.614 | 0.876 | 0.1098 |
| B. | ✗ | ✓ | ✗ | ✗ | ✗ | ✗ | ✗ | ✓ | 20.545 | 0.893 | 0.0949 | 23.366 | 0.901 | 0.0791 |
| C. | ✗ | ✗ | ✓ | ✓ | ✗ | ✗ | ✗ | ✓ | 20.167 | 0.889 | 0.1002 | 23.412 | 0.904 | 0.0794 |
| D. | ✗ | ✗ | ✓ | ✗ | ✓ | ✓ | ✗ | ✗ | 20.152 | 0.891 | 0.0976 | 23.405 | 0.903 | 0.0801 |
| E. | ✗ | ✗ | ✓ | ✗ | ✓ | ✗ | ✓ | ✗ | 19.744 | 0.886 | 0.1415 | 23.393 | 0.904 | 0.0813 |

Table 4: **Ablation Studies of our orthogonal view image generation model.**

## 4.3 ABLATION STUDIES

**Multi-View Image Generation.** In Tab. 4, we analyze our multi-view image generation model on the PuzzleIOI and 4D-Dress datasets. For feature aggregation, we compare simple averaging (A), concatenation (B), and our proposed PCFA, which achieves the best results. We also test a weighted sum strategy (C) after correlation map prediction. For reference feature extraction, we evaluate CLIP (D), DINOv2 (E), and ReferenceNet. Quantitative and visual results (Appendix F.1) show our design outperforms all alternatives.

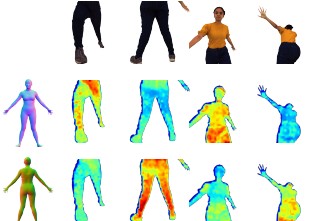

Figure 11: **Predicted Correlation Maps.** See dynamic illustration of correlation maps in *Sup.Mat.*'s *video*.

**Correlation Maps.** Our correlation map prediction module identifies and prioritizes key regions in reference images based on the target pose. As shown in Figure 11, visualizations for front- and back-view targets confirm that our maps effectively select the most relevant areas for view generation. This targeted focus improves generation quality and reduces GPU memory usage by retaining only the most informative features. More visual results are shown in Appendix E.2.

**Number of References.** In Tab. 5, quality improves as more unconstrained references are used. PCFA module efficiently selects informative features, keeping GPU memory usage low, unlike direct concatenation, which increases memory linearly.

| | Ours | | | | Concat | | | |
|---|---|---|---|---|---|---|---|---|
| | PSNR↑ | SSIM↑ | LPIPS↓ | GPU↓ | PSNR↑ | SSIM↑ | LPIPS↓ | GPU↓ |
| 3 refs | 24.159 | 0.912 | 0.0680 | 18.65 | 22.759 | 0.897 | 0.0894 | **18.02** |
| 6 refs | 25.041 | 0.917 | 0.0623 | **19.40** | 23.267 | 0.901 | 0.0807 | 24.33 |
| 9 refs | 25.646 | 0.918 | 0.0592 | **20.16** | 23.362 | 0.901 | 0.0796 | 30.89 |
| 12 refs | **25.848** | **0.920** | **0.0576** | **20.88** | 23.366 | 0.901 | 0.0791 | 37.96 |

Table 5: **Multi-View Generation with Different Number of References.**

**Robustness to Inputs & Conditions.** The generated human identity remains consistent across different target poses and reference combinations, with detailed discussions and results presented in *Sup.Mat.*'s Appendix F.2 and Appendix F.3. Notably, UP2You can effectively handle subjects with loose clothing and complex target poses, as demonstrated in Fig. 24 of Appendix F.2.

**Image Encoder of PCFA.** Given that DINOv2 has been demonstrated to effectively capture 2D-to-3D correspondences [58], we adopt it as the image encoder for our PCFA module. To further validate this design choice, we conduct additional experiments on the 4D-Dress dataset using alternative

| Inputs | Results | Inputs | Results |

Figure 12: **Generation Results on Highly Occluded Inputs.**    Figure 13: **Generation Results on Missing Part.**

image encoders, including CLIP [68] and DINOv1 [8]. As presented in Tab. 6, DINOv2 consistently outperforms both alternatives on multi-view image generation quality.

|  | PSNR↑ | SSIM↑ | LPIPS↓ |
|---|---|---|---|
| **DINOv2** | **25.848** | **0.920** | **0.0576** |
| CLIP | 23.876 | 0.904 | 0.0767 |
| DINOv1 | 24.170 | 0.907 | 0.0745 |

|  | PSNR↑ | SSIM↑ | LPIPS↓ | GPU↓ |
|---|---|---|---|---|
| $\gamma = 1.0$ | 24.978 | 0.912 | 0.0665 | **18.64** |
| $\gamma = 2.0$ | **25.848** | **0.920** | 0.0576 | 20.88 |
| $\gamma = 3.0$ | 25.837 | **0.920** | **0.0569** | 23.12 |

Table 6: **Comparison of Different Image Encoders for PCFA.**    Table 7: **Comparison of Different Number of Selected Features.**

**Number of Selected Features.** We set the default value of $\gamma$ to 2.0 to control the number of reference features selected in the `topk` selection. To determine the optimal configuration, we evaluate different values of $\gamma$ as shown in Table 7. The results demonstrate that $\gamma = 2.0$ achieves the best trade-off between generation quality and GPU memory efficiency.

**Unconstrained Inputs vs. Single Front View.** Compared to single full-body front-view inputs, unconstrained photos are easier to collect and capture richer information about side and back views. Using the comprehensive information from unconstrained photos leads to better re-

|  | PSNR↑ | | SSIM↑ | | LPIPS↓ | | Chamfer↓ | P2S↓ | Normal↓ |
|---|---|---|---|---|---|---|---|---|---|
|  | Front | Back | Front | Back | Front | Back | | | |
| Ours* | 25.257 | **25.488** | **0.906** | **0.909** | 0.0724 | 0.0733 | **1.140** | **1.119** | **0.0122** |
| PSHuman | **25.384** | 23.382 | 0.898 | 0.885 | 0.0934 | 0.1121 | 2.756 | 2.926 | 0.0189 |
| Human3Diff | 23.335 | 20720 | 0.883 | 0.872 | 0.1118 | 0.1248 | 4.275 | 4.322 | 0.0227 |
| ICON | - | - | - | - | - | - | 4.352 | 4.331 | 0.0188 |
| ECON | - | - | - | - | - | - | 3.780 | 3.642 | 0.0178 |
| PIFuHD | - | - | - | - | - | - | 2.776 | 2.603 | 0.0154 |

Table 8: **Unconstrained Photos vs. Single Front View.** * indicates our method uses unconstrained photos input, while other methods use single full-body front view input.

construction results. Table 8 compares UP2You against standard single front-view based methods on 4D-Dress dataset, including ICON [92], ECON [93], PIFuHD [72], PSHuman [48], and Human3Diff [96]. Our method achieves the best performance in both rendering quality and 3D accuracy, particularly for back-view rendering results, demonstrating the value of unconstrained inputs.

**Generated Results in Extreme Situation.** UP2You is robust to input variations and can effectively extract information from highly occluded photos. Figure 12 presents an example where one input image captures only the foot region, while other images lack this body part, demonstrating the capability of our method to handle inputs with high occlusion ratios. Figure 13 further examines scenarios where body parts are not fully visible across all images (e.g. the foot region). Due to diffusion hallucination, the generated results exhibit a somewhat reasonable structure; however, the texture is blended from other visible parts (more cases shown in *Sup.Mat.*'s Fig. 29). Therefore, inputs with complete body part coverage are more suitable for UP2You to achieve optimal results.

**Animation.** Since we adopt the A-Pose as the default target pose, the reconstructed mesh is naturally suited for animation. The textured mesh generated by UP2You can be easily animated using third-party tools such as Mixamo [6]. Moreover, the aligned SMPL-X parameters provided by UP2You enable animation based on skin weight transfer [1]. Finally, as UP2You can transform unconstrained inputs into different target pose configurations, animated rendering results can also be directly performed by itself, as demonstrated in *Sup.Mat.*'s Fig. 21.

## 5 CONCLUSION

UP2You acts as a "data rectifier," converting unconstrained photos into orthogonal views suitable for MVS. It is efficient (1.5 minutes per person on one GPU), achieves SOTA quality, and well preserves identity and clothing style across diverse input forms and pose conditions. It also enables free 3D virtual try-on (Fig. 14, more in *Sup.Mat.*'s Fig. 30). Limitations and future work are discussed in *Sup.Mat.*'s Appendix H.

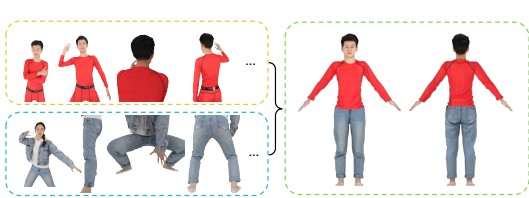

Figure 14: **3D Virtual Try-On.**

## 6 ACKNOWLEDGEMENTS

We thank *Siyuan Yu* for the help in Houdini Simulation, *Shunsuke Saito*, *Dianbing Xi*, *Yifei Zeng* for the fruitful discussions, and the members of *Endless AI Lab* for their help on data capture and discussions. This work is funded by the Research Center for Industries of the Future (RCIF) at Westlake University, the Westlake Education Foundation.

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

# Appendix

## Table of Contents

## A    USE OF LARGE LANGUAGE MODELS

We used a large language model to assist with copy editing—grammar checking, wording suggestions, and minor style and clarity improvements—after the scientific content, methodology, analyses, and conclusions had been written by the authors.

## B    RELATED WORK

### B.1    SUBJECT-DRIVEN AND ID-CONSISTENT IMAGE GENERATION

With the advent of powerful generative models [25, 42, 69, 81], subject-driven image generation has made remarkable progress in recent years. Various approaches have been proposed to generate images of specific subjects, such as optimizing specialized tokens to encode subject concepts [2, 14, 85], learning personalized modulation vectors for each concept [16], or fine-tuning pre-trained diffusion models [4, 13, 40, 70, 73] using a handful of reference images. Additionally, methods like JeDi [103] and SynCD [41] utilize global self-attention mechanisms to effectively fuse information from multiple images of a target subject, while EasyRef [109] leverages Vision-Language Models (VLMs) [5].

For human-centric generation, several methods have been developed to handle identity preservation. For instance, Omni-ID [63], IMAGPose [75], and HiFi-Portrait [95] utilize specialized image encoders to process multiple reference images for ID-preserving image synthesis. However, extending these techniques to the full body is non-trivial, as the human body's highly articulated structure and non-rigid deformations introduce significant challenges for feature fusion. To tackle this, approaches like Total Selfie [9], and RealFill [84] employ few-shot personalization via fine-tuning [70] to capture consistent identities, including both facial features and overall appearance. Nevertheless, these methods are tailored for 2D image generation and lack the mechanisms needed to ensure cross-view consistency or the precise latent feature aggregation required for high-fidelity 3D reconstruction.

## C    PRILIMINARY

We review the fundamentals of multi-view diffusion models [47, 77–79], with a particular focus on MV-Adapter [32], which serves as the foundation for the multi-view generation of UP2You.

**Multi-View Diffusion Models.** Multi-view diffusion models extend single-view generation by introducing multi-view attention mechanisms, enabling the synthesis of images that are consistent across different viewpoints. Several works [78, 79] generalize the self-attention mechanism of standard diffusion models to operate over all pixels from multiple views. Specifically, given $f^{\text{in}}$ as the input to the attention block, multi-view self-attention concatenates features from $M$ views, allowing the model to capture global dependencies. However, this approach incurs significant computational overhead due to the need to process all pixels across all views. To mitigate this, row-wise self-attention [47, 48] leverages geometric correspondences between orthogonal views. For example, Era3D [47] restricts attention to the current view and corresponding rows from other views, which is well-suited for orthogonal multi-view generation and substantially reduces computational cost.

Building on row-wise self-attention, MV-Adapter [32] introduces an image-to-multiview (I2MV) generator with a parallel attention architecture. The original self-attention block is modified as:

$$f^{\text{self}} = \texttt{SelfAttn}(f^{\text{in}}) + \texttt{MVAttn}(f^{\text{in}}) + \texttt{RefAttn}(f^{\text{in}}, \mathbf{F}) + f^{\text{in}}, \qquad (5)$$

Here, `MVAttn` represents the row-wise self-attention mechanism, while `RefAttn` is a cross-attention module that integrates the reference image feature $\mathbf{F}$ into $f^{\text{in}}$. The feature $\mathbf{F}$ is extracted from the input image $\mathbf{I}$ using the reference network $\mathcal{R}$ [27]: $\mathbf{F} = \mathcal{R}(\mathbf{I})$. The I2MV generation process in MV-Adapter is formulated as $\mathbf{V} = \mathcal{D}(\mathbf{F}, \mathcal{P}(\mathbf{P}))$, where $\mathbf{V} = \{\mathbf{V}_1, \mathbf{V}_2, \ldots, \mathbf{V}_M\}$ denotes the set of generated multi-view images, $\mathcal{D}$ represents the multi-view diffusion model, $\mathbf{P} = \{\mathbf{P}_1, \mathbf{P}_2, \ldots, \mathbf{P}_M\}$ specifies the target viewpoint conditions, and $\mathcal{P}$ is the condition encoder that fuses viewpoint conditions into $\mathcal{D}$. In MV-Adapter, only `MVAttn`, `RefAttn`, and $\mathcal{P}$ are trained for I2MV generation. Each $\mathbf{P}$ is encoded as a camera ray representation, referred to as a "raymap". Typically, $M = 6$ orthogonal views are generated, corresponding to the target view angles $\{0°, 45°, 90°, 135°, 180°, 270°\}$.

Given the efficient plug-and-play adapter training mechanism of MV-Adapter, combined with the robust feature extraction capabilities of ReferenceNet for processing unconstrained photographs, we adopt MV-Adapter as our multi-view diffusion model architecture. Furthermore, considering our focus on human-centric tasks, we utilize SMPL-X normal rendering as the viewpoint condition $\mathbf{P}$.

# D    IMPLEMENTATION DETAILS

## D.1    MODEL STRUCTURE

We adopt the framework architecture of MV-Adapter [32] with the `stable-diffusion-2-1-base` version [82] as the foundation for both multi-view image and normal generation. The number of selected reference features $\gamma$ is set to 2.0 during both training and inference phases. We employ the `DINOv2-Large` [11] variant of the DINOv2 encoder $\mathcal{E}^{\text{ref}}$. For the pose image encoder $\mathcal{E}^{\text{ref}}$, we implement a lightweight ResNet [22] architecture. The learnable shape tokens $\tau \in \mathbb{R}^{10 \times 1024}$ are configured to align with the dimensions of $\mathcal{E}^{\text{ref}}$, and the perceiver blocks in $\mathcal{S}$ comprise 6 layers of cross-attention.

## D.2    DATASET

We train our multi-view image generation, normal map generation, and shape prediction models using the THuman2.1 [99], Human4DiT [74], 2K2K [20], and CustomHumans [24] datasets. Since our task requires handling scenarios where individuals with the same identity appear in different poses, we manually filter the data and group samples by identity. The final training dataset comprises 6,921 scans spanning 2,091 distinct identities. For each scan, we render 6 orthogonal views ($\{0°, 45°, 90°, 135°, 180°, 270°\}$) of both images and normal maps, along with the corresponding SMPL-X normal rendering. Additionally, we render 8 views of each scan using randomly selected perspective cameras to provide "unconstrained photos". During orthogonal image generation training, for each case, we randomly select 3 to 8 reference images from other cases sharing the same identity.

For evaluation, we select 40 identities from PuzzleIOI [94] and additionally choose "A-pose" configurations from all 68 identities in 4D-Dress [88], while utilizing the remaining poses as reference views. To ensure that SMPL-X camera normal rendering accurately represents viewpoint information, we rotate all scans so that the front view corresponds to zero azimuth. Beyond synthetic data, we also collect an in-the-wild dataset comprising 12 identities for further evaluation, ensuring robust evaluation in diverse scenarios.

## D.3    TRAINING DETAILS

We train the image and normal generation models end-to-end using denoising losses $\mathcal{L}_d^{\text{rgb}}$ and $\mathcal{L}_d^{\text{normal}}$, respectively. During training, $\mathcal{L}_d^{\text{rgb}}$ jointly optimizes the components $\mathcal{E}^{\text{pose}}$, $\mathcal{T}$, $\mathcal{W}_q$, $\mathcal{W}_k$, `AvgPool`, $\mathcal{P}^{\text{rgb}}$, and $\mathcal{D}^{\text{rgb}}$. In normal maps generation training, $\mathcal{L}_d^{\text{normal}}$ optimizes $\mathcal{P}^{\text{normal}}$ and $\mathcal{D}^{\text{normal}}$. For shape prediction, we employ the loss function $\mathcal{L}_v = |\mathbf{T}(\beta^{\text{pred}}) - \mathbf{T}(\beta^{\text{gt}})|$ to compute the vertex-wise distance between SMPL-X meshes generated from the predicted shape parameters $\beta^{\text{pred}}$ and the ground-truth shape parameters $\beta^{\text{gt}}$.

The complete training process for the image and normal generation models requires approximately 3 and 2 days, respectively, on 8 NVIDIA 5880 GPUs. We employ a batch size of 1 per GPU under bfloat16 mixed precision and train for 50,000 iterations. All pose, input, and output image resolutions are consistently set to $768 \times 768$. The reference images for both image and normal generation are also configured at $768 \times 768$ resolution, while the target orthogonal view angles follow the same configuration as MV-Adapter. The shape prediction model undergoes training for 100,000 iterations on 8 NVIDIA 5880 GPUs with a batch size of 8 per GPU, requiring approximately 10 hours. We apply a constant learning rate of $5 \times 10^{-5}$ with warm-up for training all models.

## D.4    EVALUATION METRICS

We employ three complementary metrics to assess geometric accuracy: (1) **Chamfer distance** (bidirectional point-to-surface distance in cm), which measures overall geometric similarity; (2) **P2S**

**distance** (unidirectional point-to-surface distance in cm), which captures reconstruction completeness; and (3) **L2 error for Normal maps** rendered from four canonical views ($\{0°, 90°, 180°, 270°\}$), which evaluates fine-grained surface detail preservation.

We render multi-view color images from the same four canonical viewpoints and evaluate appearance fidelity using three established image quality metrics: **PSNR** (Peak Signal-to-Noise Ratio) for pixel-level accuracy, **SSIM** (Structural Similarity) for structural consistency, and **LPIPS** (Learned Perceptual Image Patch Similarity) for perceptual similarity.

For the in-the-wild dataset, which lacks 3D ground truth, we assess reconstruction quality using perceptual similarity metrics **CLIP-I** and **DINO** computed between the generated front view and the captured reference front view image with A-pose.

We further evaluate shape prediction accuracy by computing vertex-to-vertex (V2V) distances between predicted and ground truth SMPL-X meshes under canonical T-pose (zero pose and expression).

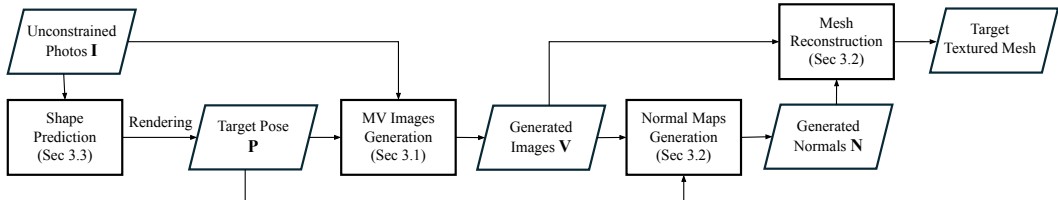

Figure 15: **Inference Process of UP2You.** Given only unconstrained photos **I** as inputs, UP2You can generate a high-quality textured mesh.

## D.5 INFERENCE PROCESS

The inference process of UP2You for unconstrained photo inputs **I** is illustrated in Fig. 15, which mainly consists of four steps as follows:

(1) Use $\mathcal{S}$ to estimate SMPL-X shape parameters $\beta^{\text{pred}}$ from **I**, and initialize the SMPL-X mesh with $\beta^{\text{pred}}$ and a predefined pose (e.g., A-pose with zero expression) to obtain the pose condition **P**.

(2) Generate multi-view images **V** using $\mathcal{D}^{\text{rgb}}$, conditioned on **I** and **P**.

(3) Generate multi-view normal maps **N** using $\mathcal{D}^{\text{normal}}$, conditioned on **V** and **P**.

(4) Reconstruct the textured mesh using the initialized SMPL-X mesh, **V**, and **N**.

For data pre- and post-processing, we employ [106] to remove backgrounds from input unconstrained photos. Additionally, the reference masks are resized and adapted to the correlation maps **C** to enhance the model's focus on foreground regions.

**Inference Time.** The complete pipeline requires approximately 1.5 minutes to generate a textured mesh from a single unconstrained input. Specifically, the shape prediction step takes about 1 second, multi-view image generation requires approximately 15 seconds, normal map generation takes about 15 seconds, and mesh reconstruction, along with other processing steps (e.g., foreground segmentation, data postprocessing, and file saving), takes nearly 1 minute.

# E ADDITIONAL VISUAL COMPARISONS

## E.1 QUALITATIVE COMPARISONS

We present additional qualitative comparison results in Figs. 16 to 19, including mesh reconstruction, front-view 3D human reconstruction, and shape prediction comparisons. Please zoom in for details.

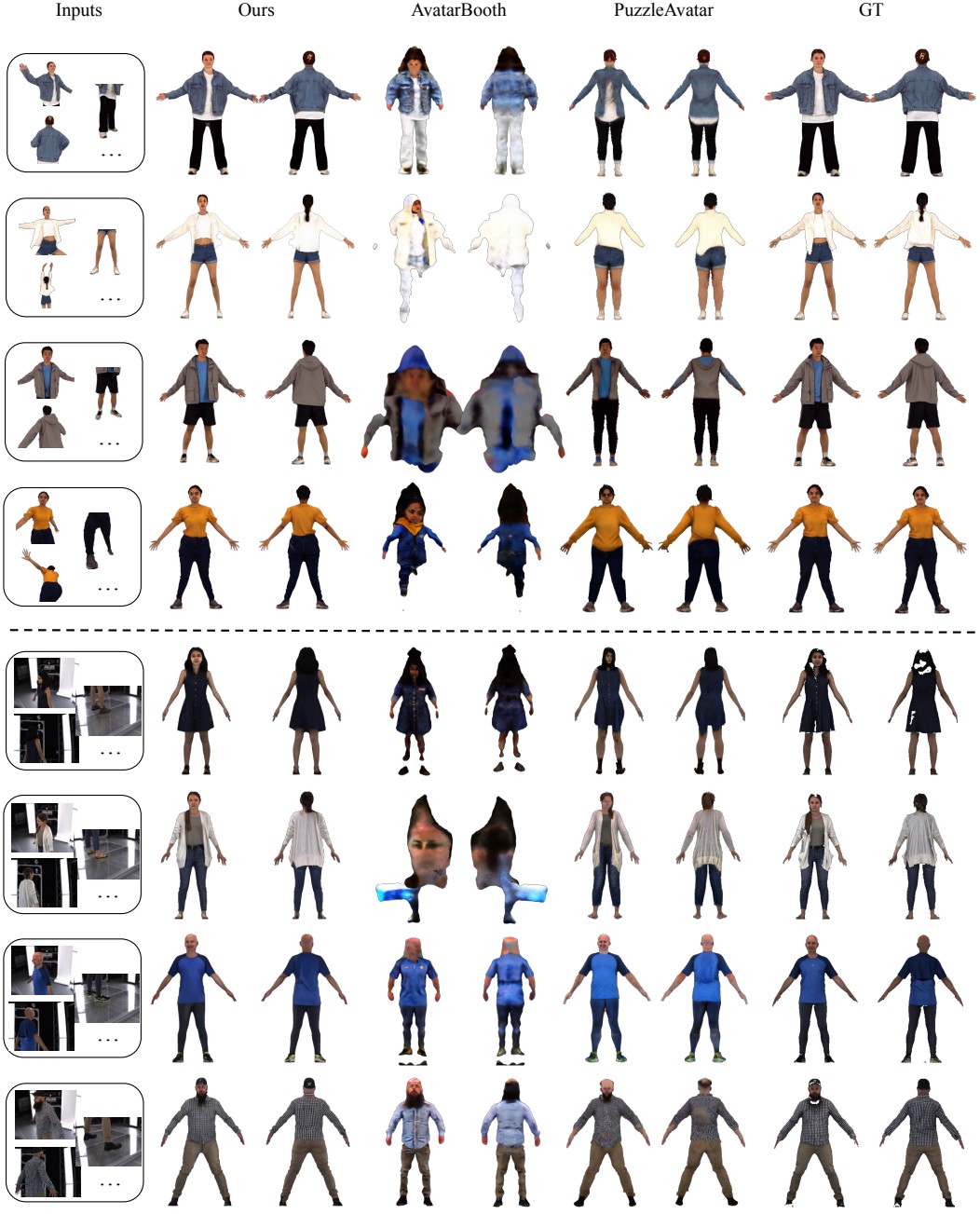

Figure 16: **More Qualitative Comparisons on 4D-Dress and PuzzleIOI datasets.**

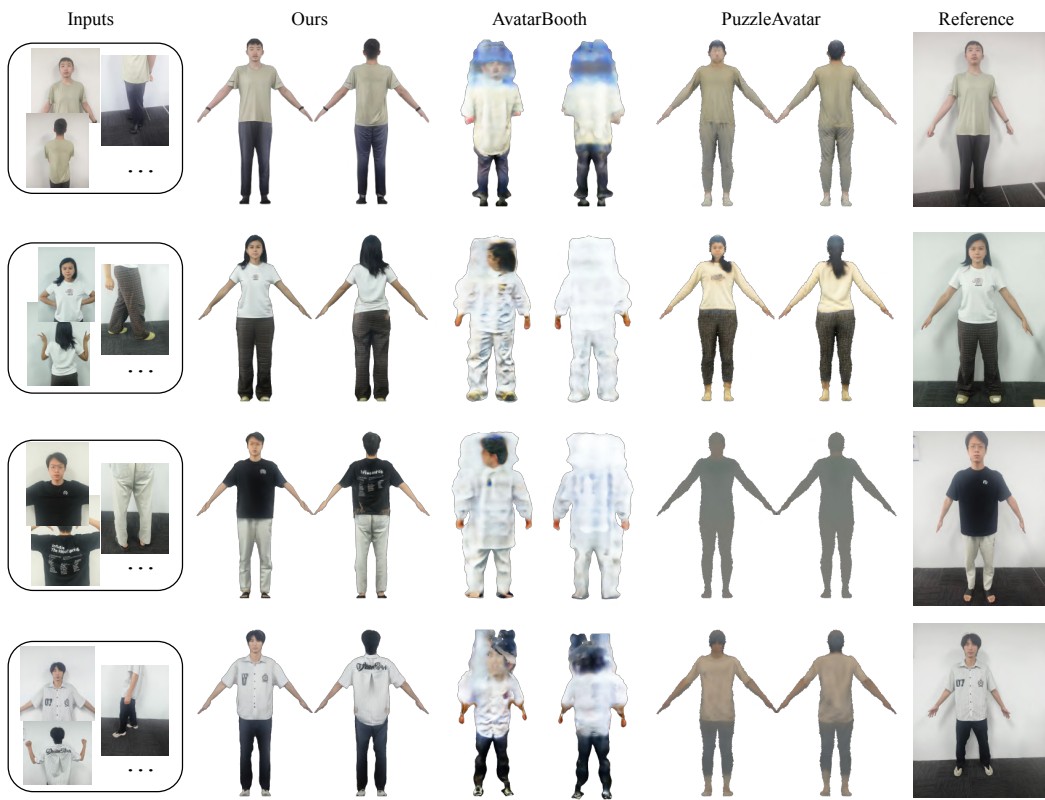

Figure 17: **More Qualitative Comparisons on in-the-wild dataset.**

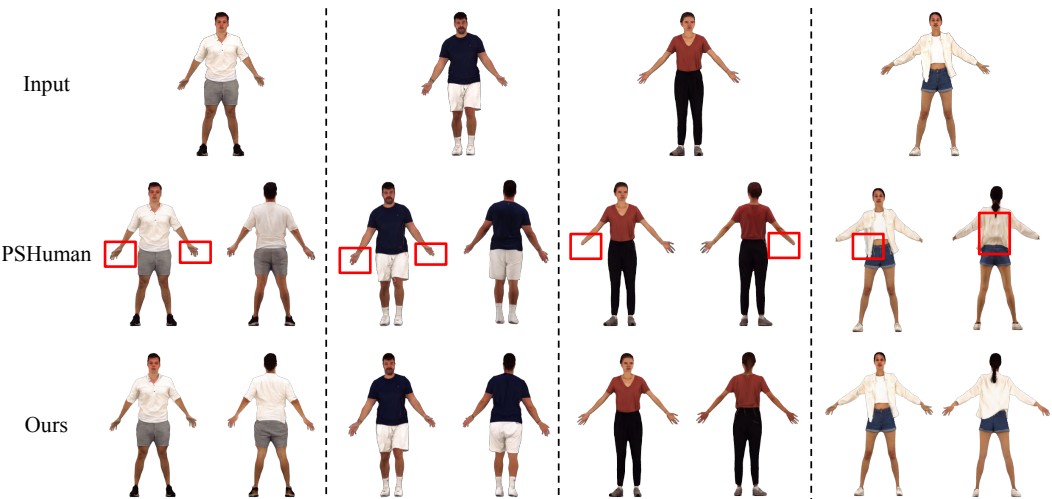

Figure 18: **More Qualitative Comparisons of Single Image 3D Human Reconstruction with PSHuman.**

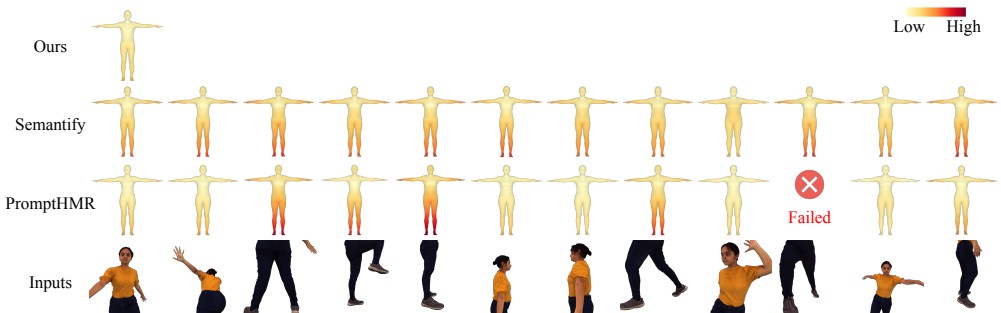

Figure 19: **Error Maps of Shape Prediction.**

### E.2 CORRELATION MAPS

Pose-dependent correlation maps generation is an important module of UP2You, as the first part of the proposed PCFA, it predicts the most relevant regions of input unconstrained photos for the conditioned pose. With the latter feature selection strategy, PCFA can focus on informative features for viewpoint generation. In Fig. 20, we provide more results of the generated correlation maps.

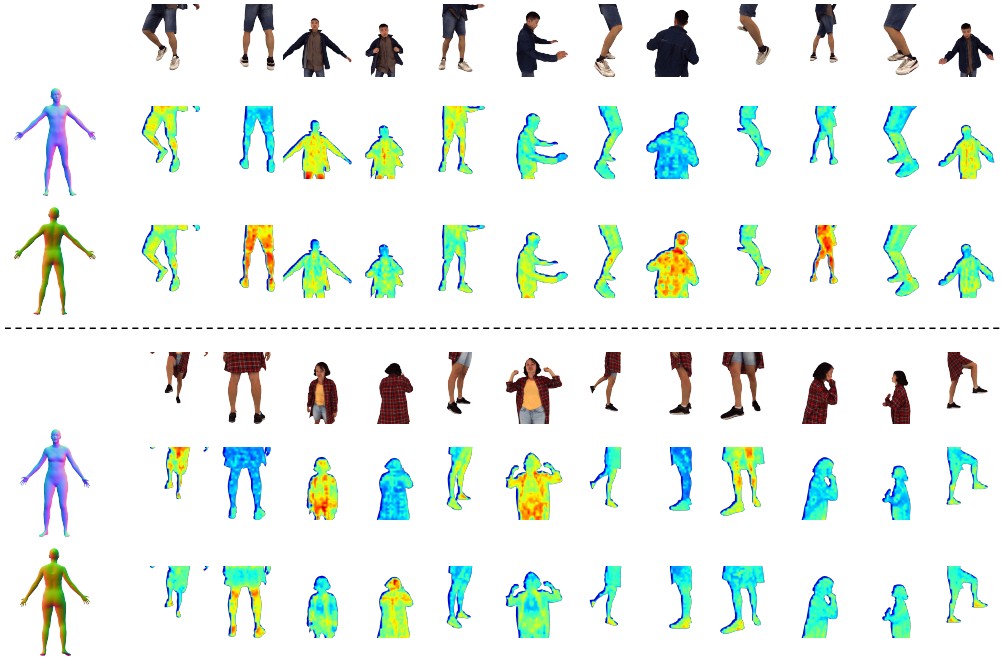

Figure 20: **Visualize Results of Correlation Maps.** Given the input reference images and target pose for multi-view image generation, the predicted correlation maps can effectively identify and discriminate correlated regions within the reference inputs. For example, when generating images in the front-view, reference regions that correspond to front-facing views exhibit higher correlation values, demonstrating the model's ability to selectively attend to relevant spatial information.

### E.3 ANIMATION RESULTS

We present an animation sample generated by UP2You using the same reference with different target poses, as shown in Fig. 21. Notably, UP2You maintains identity consistency well across different target poses. However, since this approach just reconstructs a textured mesh independently for each frame, temporal consistency of the rendered images and mesh topology is not guaranteed. For production-quality animated sequences, we recommend using professional animation methods and tools [1, 6] for textured mesh animation.

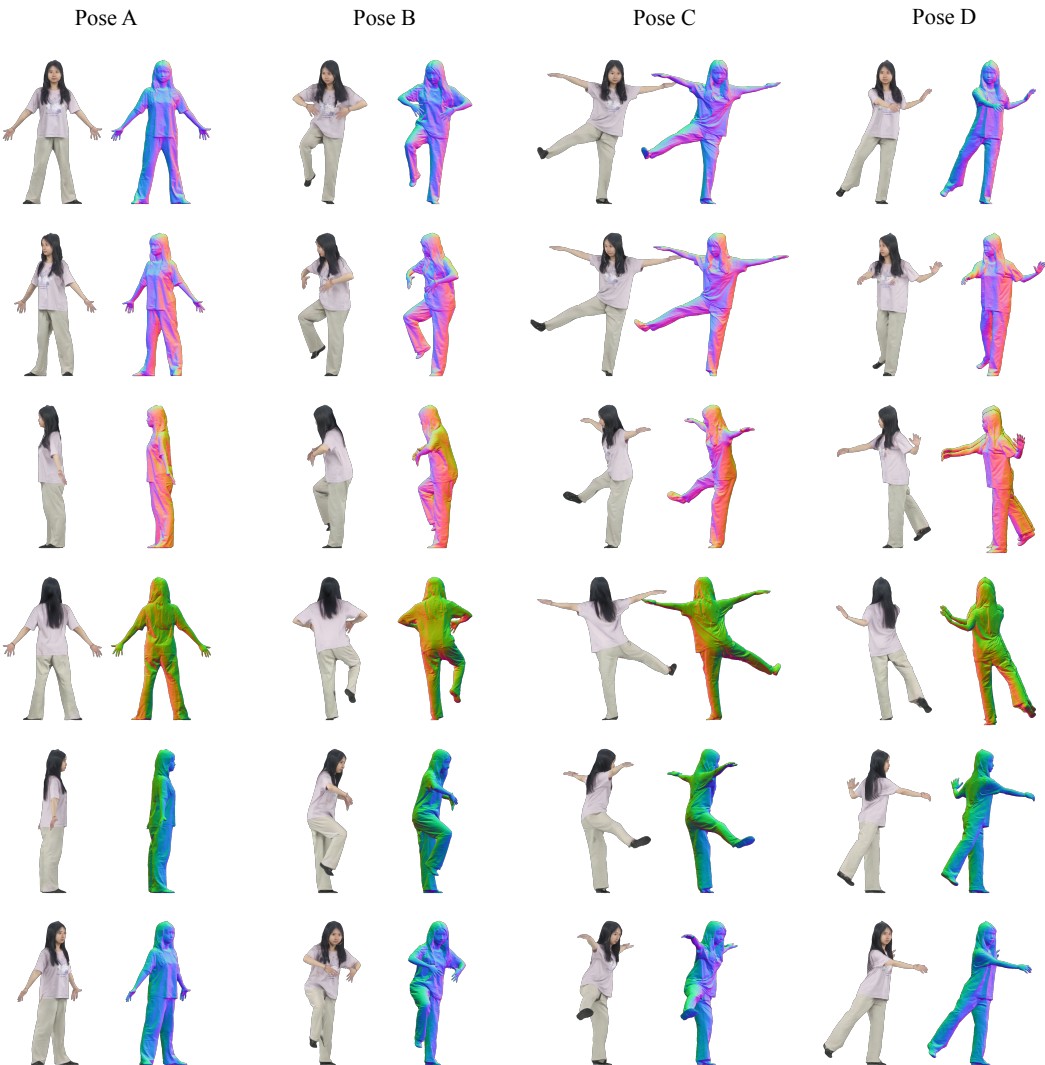

Figure 21: **Animation Results of Textured Mesh Generated by UP2You.**

# F    ADDITIONAL ABLATION STUDIES

## F.1    VISUAL RESULTS OF DIFFERENT ORTHOGONAL IMAGES GENERATION DESIGNS

Here, we present the generated visual results in Fig. 22 for different design choices in the multi-view image generation model. As indicated earlier, approach A directly concatenates all reference features for viewpoint generation, which may provide irrelevant features during generation and lead to poor results. Approach B averages all reference features as global guidance. This method is time-efficient but loses important color features and generates suboptimal results. Approach C uses a weighted sum strategy to aggregate reference features after computing the correlation map, which loses details in some regions since regions with high correlation values may overlap. Approaches D and E utilize CLIP and DINOv2 features, respectively, rather than ReferenceNet as in our method. CLIP features have low resolution and are difficult to preserve details such as facial and clothing textures, while DINOv2 is texture-insensitive and thus difficult to restore reference textures accurately.

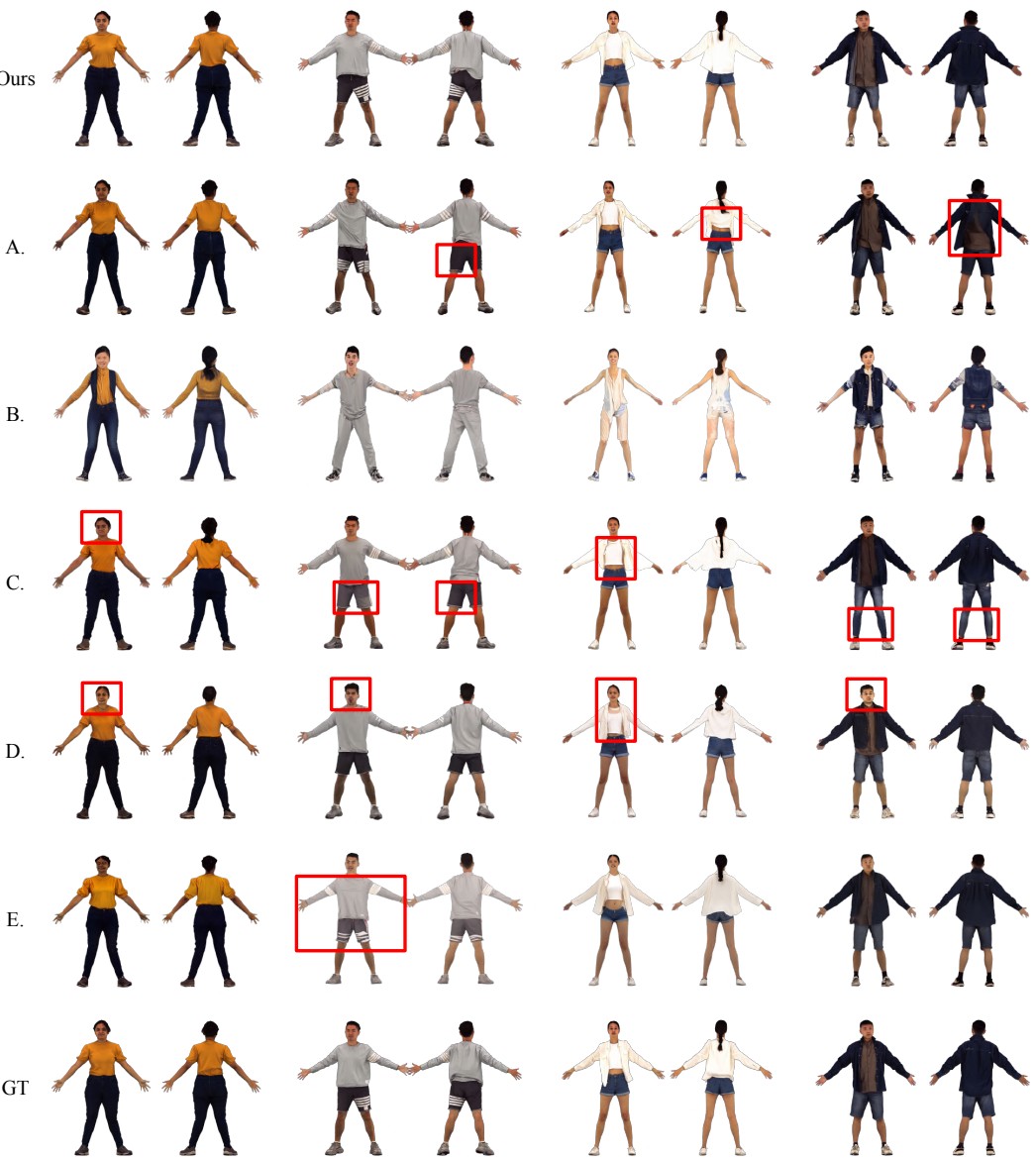

Figure 22: **Visual Comparisons of Different Multi-View Image Generation Designs.**

## F.2 ROBUSTNESS OF TARGET POSE CONDITION.

While previous experiments highlight the strong generation ability of UP2You, most target poses are in the "A-pose" configuration. Since 4D-Dress provides ground-truth multi-view images of persons with different poses, we further test robustness by randomly selecting three diverse target poses per identity from the 4D-Dress dataset and evaluating our multi-view image generation performance. As shown in Tab. 9, UP2You maintains high-quality results across varied target poses using the same unconstrained photo inputs. Figure 23 further demonstrates the visual results, where identity is consistently preserved across different poses. In addition, Figure 24 shows the generation results on subjects with loose clothing and complex target poses, further validating the generation capability and robustness of UP2You.

|  | PSNR↑ | SSIM↑ | LIPIPS↓ |
|---|---|---|---|
| Pose A | 24.983 | 0.911 | 0.0664 |
| Pose B | 24.400 | 0.902 | 0.0744 |
| Pose C | 24.519 | 0.904 | 0.0715 |

Table 9: **ID Consistency.** UP2You achieves high-quality multi-view image generation results in 4D-Dress dataset in three different pose condition.

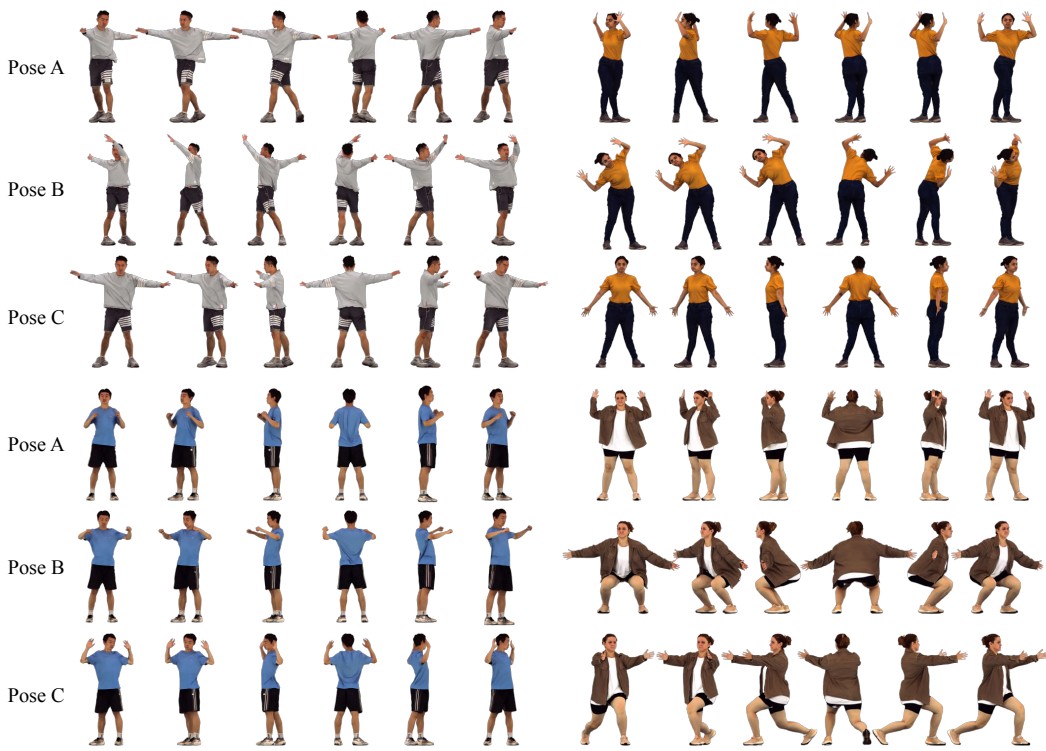

Figure 23: **Robustness of target pose conditions.** Our method can generate high-quality multi-view images under different pose conditions with the same reference inputs, demonstrating that identity information is effectively disentangled from pose conditions in our approach.

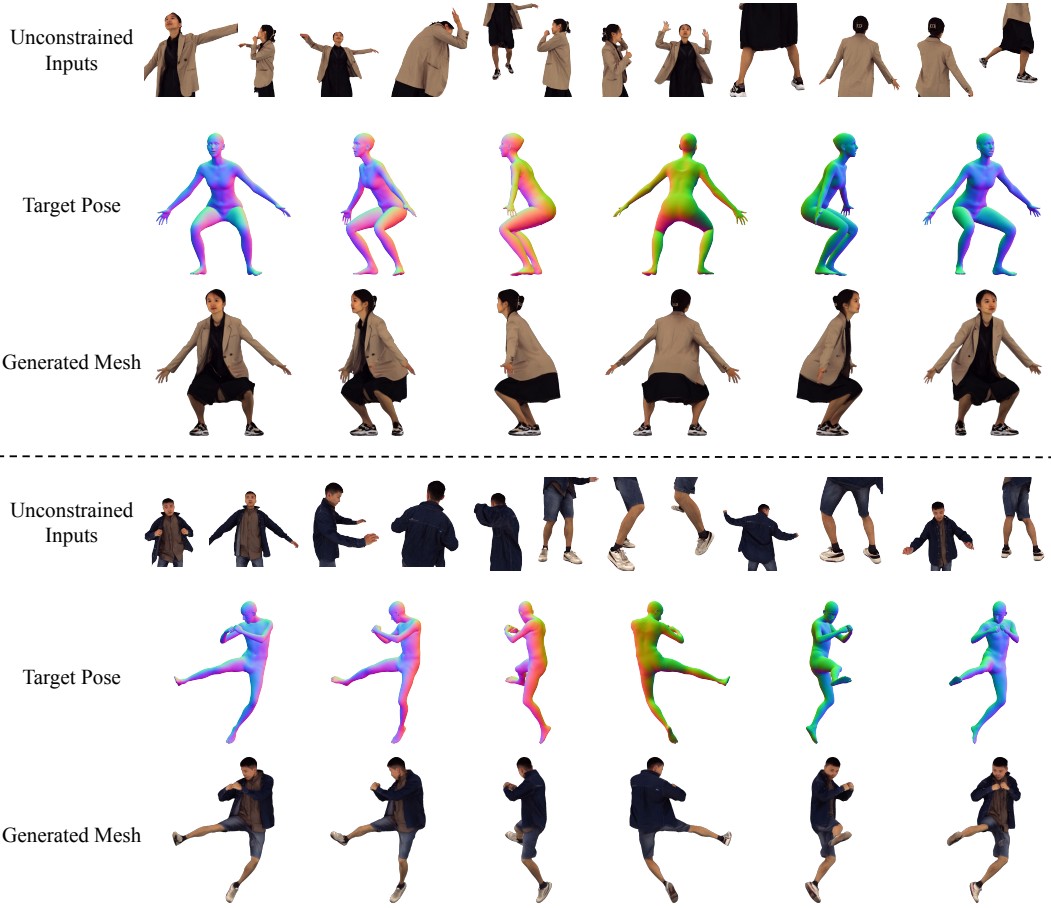

Figure 24: **Generation Results with Loose Clothing and Complex Target Pose.**

### F.3 ANALYSIS OF SHAPE PREDICTOR.

To evaluate whether our shape predictor can regress consistent shape parameters, we assess our shape prediction model using different groups of unconstrained reference inputs from the same identity. As shown in Tab. 10, our method achieves stable shape predictions across all input groups. Since the aggregated pixel-level features from reference inputs may contain information about personal shape characteristics, the multi-view image generation model in UP2You exhibits some degree of robustness to shape variations. However, in extreme cases, more accurate shape predictions can significantly enhance the quality

|  | Ref Group A | Ref Group B | Ref Group C |
|---|---|---|---|
| V2V↓ (mm) | 7.485 | 7.503 | 7.443 |

Table 10: **Shape prediction consistency on the 4D-Dress dataset.** We input three different groups of 12 reference images of the same person into our shape predictor. The vertex-to-vertex (V2V) error of the predicted results shows stable values with low variance, demonstrating that our shape predictor is robust to unconstrained reference inputs.

of the final 3D human generation. We evaluate the impact of our shape predictor on the overall inference pipeline of UP2You and find that incorporating the proposed shape predictor leads to measurable improvements in generation quality on the in-the-wild dataset. As demonstrated in Fig. 25, our shape predictor enables more identity-consistent results for individuals with extreme body shapes, while Tab. 11 provides quantitative evidence that the proposed shape predictor improves performance on the in-the-wild dataset.

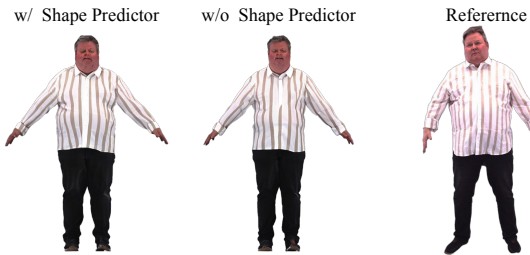

w/ Shape Predictor     w/o Shape Predictor     Referernce

|  | w/ Shape Predictor | | w/o Shape Predictor | |
|---|---|---|---|---|
|  | Ours Image | Ours Mesh | Ours Image | Ours Mesh |
| CLIP-I↑ | **0.972** | 0.971 | 0.969 | 0.969 |
| DINO↑ | **0.932** | 0.916 | 0.927 | 0.911 |

Table 11: **Effects of Shape Predictor on the in-the-wild Dataset.** Generation results with the aid of shape predictor have better performance.

Figure 25: **Shape Predictor Helps to Generate More Identity-Consistent Results for People in Extreme Shape.**

## F.4 VISUAL RESULTS WITH DIFFERENT NUMBER OF INPUTS

In UP2You, as more unconstrained photos are provided as input, additional details can be extracted and refined in orthogonal views, thereby improving the reliability of the generated results. We demonstrate this principle through an illustrative example in Fig. 26.

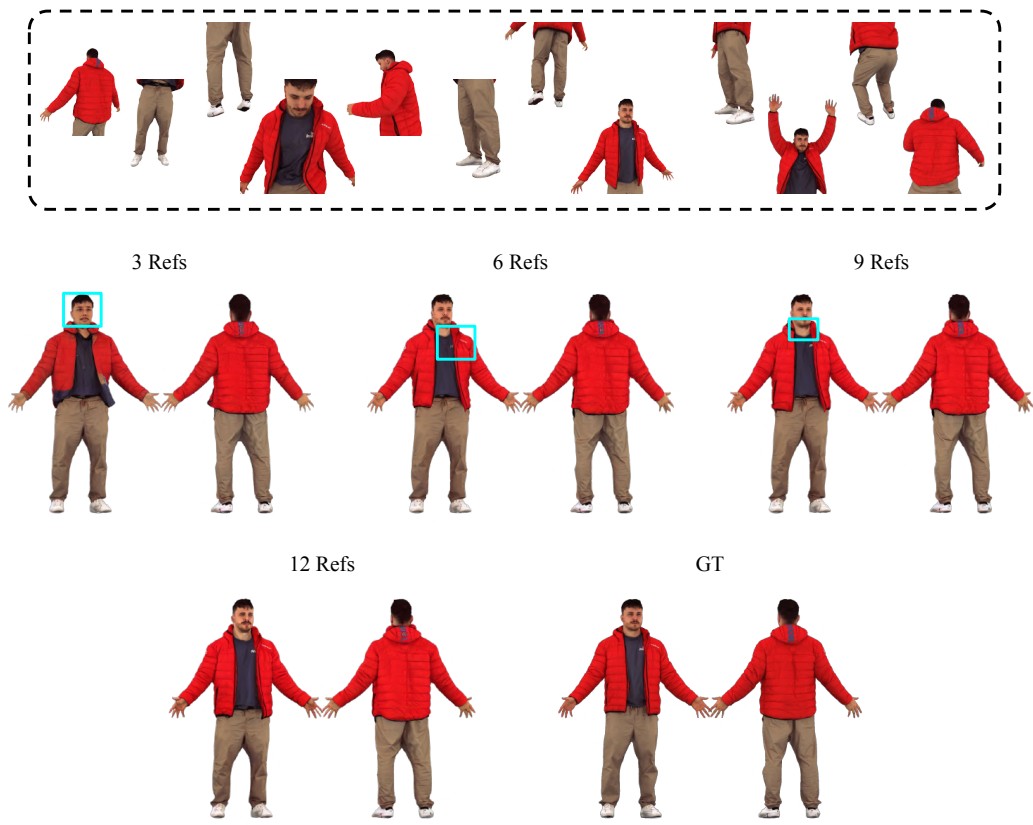

Figure 26: **Generated Multi-View Image Results with Different Number of References.** With more references input, more results are noticed and generated by our model, like facial details and clothing patterns.

# G MORE GENERATION RESULTS OF UP2YOU

Figures 27 and 28 present comprehensive generation results of UP2You on two representative cases, including the reference images, generated multi-view images and normal maps, as well as the rendered images and normal maps after mesh reconstruction. Figure 29 demonstrates that UP2You is robust to diverse inputs, performing well even in extreme scenarios such as inputs missing the face, lower body, or upper body. Additionally, Figure 30 provides further examples of 3D virtual try-on applications.

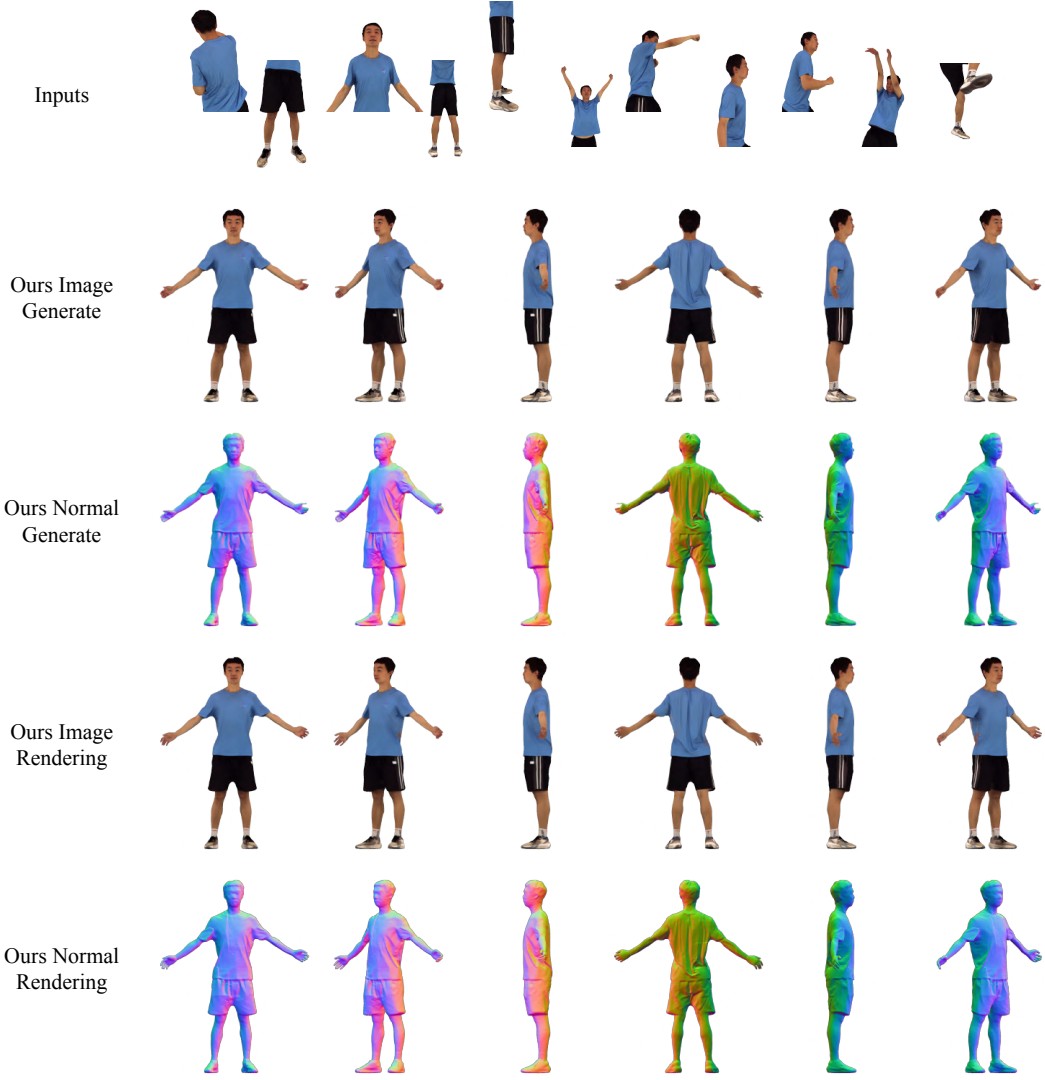

Figure 27: **Generated Results of UP2You.**

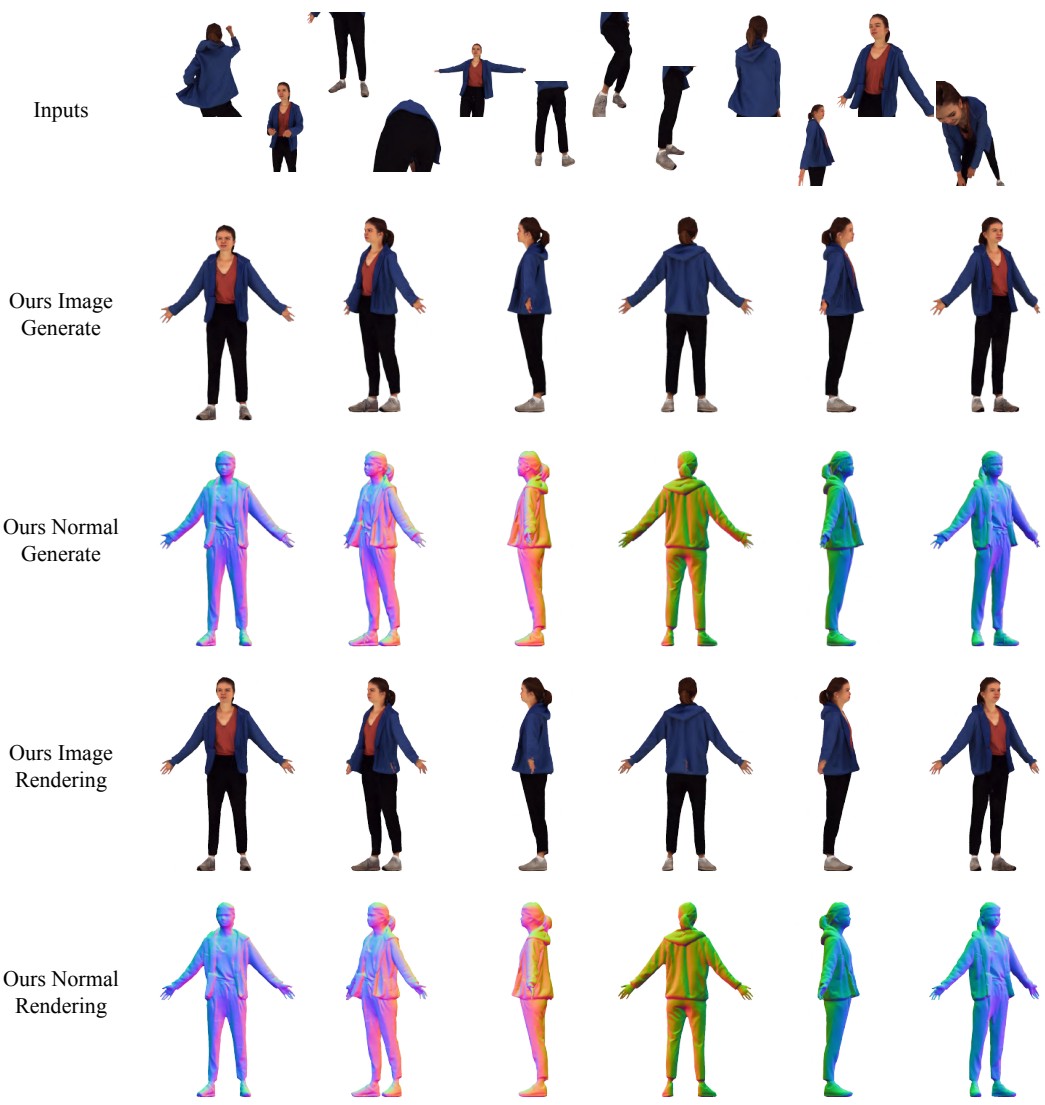

Figure 28: **Generated Results of UP2You.**

Inputs                                    Results

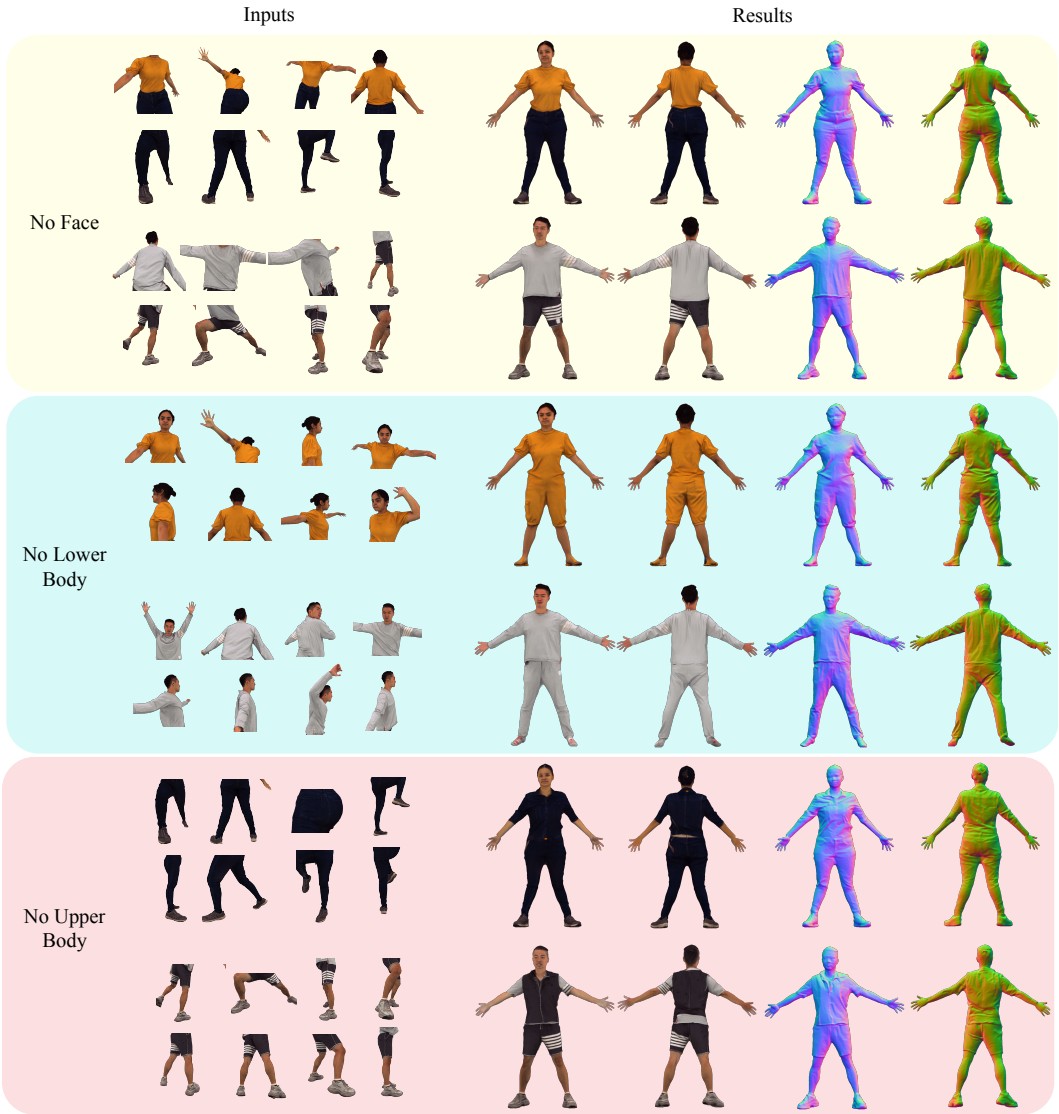

Figure 29: **More Generation Cases with Invisible Parts.** UP2You generates reasonable results with different kinds of invisible scenarios.

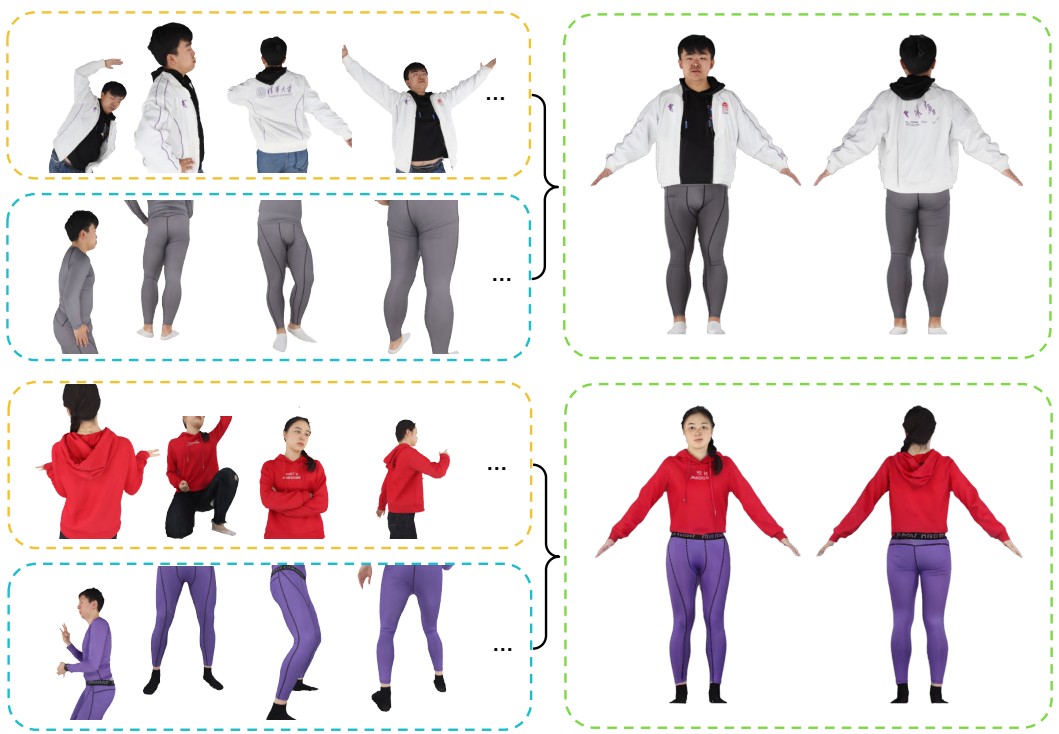

Figure 30: **More examples of 3D Virtual Try-On.**

# H    LIMITATIONS AND FUTURE WORKS

While our method shows promising results in generating high-quality 3D human avatars from unconstrained photos, there are still some limitations that we plan to address in future work:

- **Dependence on 3D Data for Training:** Our method relies on a dataset of 3D human models for training the diffusion model. Acquiring high-quality 3D data can be challenging and may limit the diversity of the generated avatars. In future work, we aim to explore semi-supervised or unsupervised approaches that can leverage large-scale 2D image or video datasets to reduce this dependence on 3D data.

- **Texture Misalignment:** Our method generates 6 orthogonal views for mesh reconstruction and texturing, which is insufficient for high-quality texture baking. Texture misalignment issues may arise in some cases (Fig. 31). In future work, we plan to adopt video generation models as the base framework for dense view synthesis to address this limitation.

- **Multiple Inference Stages:** When processing in-the-wild photos, our mesh reconstruction pipeline involves four sequential stages: shape prediction, multi-view image generation, multi-view normal map generation, and mesh reconstruction. This multi-stage inference approach slows down the generation process and may introduce cumulative errors. We plan to develop a feed-forward model that directly predicts the final results.

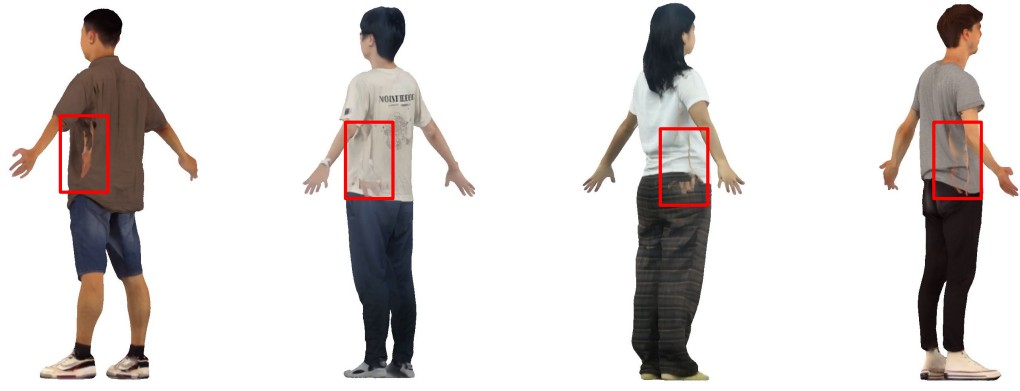

Figure 31: **Failure Cases of UP2You.** Since only 6 orthogonal views {0°, 45°, 90°, 135°, 180°, 270°} are generated, the backside texture of generated humans is lacking in guidance, making the problem of texture misalignment.

