# OpenReview forum: "UP2You: Fast Reconstruction of Yourself from Unconstrained Photo Collections"
_ICLR.cc/2026/Conference — ICLR 2026 Poster_

### Official Review · Reviewer_VnCH · 2025-10-27

**Soundness:** 3
**Presentation:** 3
**Contribution:** 2
**Rating:** 4
**Confidence:** 3

**Summary:**

The paper introduces UP2You, a tuning-free method for reconstructing high-fidelity 3D textured meshes of clothed humans from unconstrained photo collections (e.g., partial views, occlusions, varying poses, and viewpoints). Unlike prior approaches that rely on personalization via fine-tuning (e.g., DreamBooth + SDS optimization, which take hours), UP2You adopts a "data rectifier" paradigm: it aggregates features from input photos using a Pose-Correlated Feature Aggregation (PCFA) module within an MV-Adapter diffusion backbone to generate clean orthogonal multi-view RGB images and normals in seconds. It also includes a perceiver-based multi-reference shape predictor to estimate SMPL-X parameters without ground-truth templates. The pipeline culminates in mesh carving and texture baking adapted from PSHuman.

**Strengths:**

1. The tuning-free design enables fast reconstruction (1.5 minutes total, with rectification in <15 seconds), scaling well to varying numbers of inputs (1 to dozens) with near-constant memory via PCFA's selective aggregation. This makes it suitable for real-world scenarios like personal photo albums, outperforming slow optimization-based methods like PuzzleAvatar (4+ hours).
2. PCFA effectively fuses pose-correlated features from diverse photos, leading to better identity preservation and 3D consistency. Quantitative gains over SOTAs (e.g., PuzzleAvatar, AvatarBooth, PSHuman) in geometry and texture metrics, plus qualitative scaling with more inputs, demonstrate robustness to occlusions, croppings, and viewpoints.
3. The perceiver-based shape predictor eliminates the need for pre-captured templates or ground-truth shapes, enabling flexible applications like random pose animation and multi-garment virtual try-on—all without additional training.

**Weaknesses:**

1. Limited core innovation in the pipeline: The method heavily builds on existing components (e.g., MV-Adapter for diffusion-based multi-view generation, PSHuman for final mesh reconstruction), with novelty primarily in PCFA and the shape predictor. This feels like a serial integration (rectification via finetuned diffusion + off-the-shelf reconstruction), potentially lacking a unified, groundbreaking advance beyond combining A+B-style modules.
2. Dependency on diffusion priors and potential hallucinations: As a diffusion-based approach, it inherits risks of inconsistent outputs in extreme cases (e.g., highly occluded or sparse inputs), and the paper does not deeply address failure modes or ablation on the limits of rectification without further personalization.
3. Evaluation scope and generalizability: Benchmarks are limited to specific datasets (PuzzleIOI, 4D-Dress, in-the-wild captures); lacks broader comparisons to recent diffusion-free methods or real-time systems. Runtime claims (1.5 minutes) may not hold on lower-end hardware, and virtual try-on is demonstrated but not quantitatively evaluated against specialized try-on models.

**Questions:**

1. Could you elaborate on the novelty of PCFA beyond standard attention mechanisms in diffusion models? How does it differ from similar feature selection in prior works like ReferenceNet or multi-view adapters, and what ablations justify its design over simpler top-k strategies without pose correlation?
2. The pipeline relies on MV-Adapter finetuning for rectification and PSHuman for reconstruction—how much of the performance gains stem from these existing tools versus UP2You's contributions? Would the method work with alternative backbones, or is it tightly coupled to diffusion-based generation?
3. For the shape predictor, how robust is it to very sparse inputs (e.g., 1-2 photos with heavy occlusions)? The paper mentions consistency across inputs, but could you provide metrics on shape estimation variance or comparisons to single-image HMR methods like BEDLAM or HYBRIK?
4. The weakness in innovation seems tied to the serial A+B nature (diffusion rectification + existing recon); how does UP2You advance beyond a straightforward combination of MVDiffusion and PSHuman, especially since normals/meshes are generated using adapted priors from these?

---

> ### Author Response · Authors · 2025-11-21
> **Official Response by Authors -- Part 1**
>
> We would like to sincerely thank the reviewer for the detailed and constructive comments on our work. We take every comment seriously and hope our response can address the reviewer’s concerns. If there are any remaining questions, we are more than happy to address them.
>
> > **Q1: Limited core innovation in the pipeline: The method heavily builds on existing components (e.g., MV-Adapter for diffusion-based multi-view generation, PSHuman for final mesh reconstruction), with novelty primarily in PCFA and the shape predictor. This feels like a serial integration (rectification via finetuned diffusion + off-the-shelf reconstruction), potentially lacking a unified, groundbreaking advance beyond combining A+B-style modules.**
>
> **TL;DR: UP2You's core innovation is a "rectify-and-reconstruct" paradigm featuring two key components: the orthogonal view rectifier (w/ PCFA) and the shape predictor. Unlike PuzzleAvatar, the first to address unconstrained human photos, UP2You 1) does not require GT body for shape initialization, 2) scales well with more inputs, and 3) is much faster (1.5 minutes vs. 4 hours). All of the above stem from our novel paradigm.**
>
> While multi-view diffusion models (PSHuman, MV-Adapter) are effective for single-view 3D generation, they struggle with unconstrained inputs. Our rectifier design overcomes this by adapting these methods to handle unstructured photos—a non-trivial contribution.
>
> The PCFA module with top-k selection chooses the most informative spatial features from unconstrained photos for each target view generation, whose novelty has been validated by reviewers FySx and zhP9. The shape predictor leverages multiple inputs to estimate body shape parameters, outperforming existing methods (Table 2).
>
> These designs are purpose-built architectures, not simply fine-tuned diffusion models, solving the previously unsolved challenge of efficiently reconstructing humans from unconstrained photos (PuzzleAvatar needs 4-5 hours).
>
> The differentiable reconstruction backend of PSHuman is continuous remeshing [1], which is modular and can be replaced with many other methods, e.g. NeuS, 2DGS. We don't claim this is the contribution of UP2You.
>
> [1] Palfinger, Werner. "Continuous remeshing for inverse rendering." Computer Animation and Virtual Worlds 33.5 (2022): e2101.
>
> ---
>
> > **Q2: The weakness in innovation seems tied to the serial A+B nature (diffusion rectification + existing recon); how does UP2You advance beyond a straightforward combination of MVDiffusion and PSHuman, especially since normals/meshes are generated using adapted priors from these?**
>
> **TL;DR: MVDiffusion model and PSHuman are designed for single-view input and cannot directly handle multiple unconstrained inputs, which is the core challenge that UP2You addresses.**
>
> As discussed in Q1, despite using diffusion priors and differentiable 3D optimization, UP2You introduces a novel data rectification pipeline that extends beyond simple combination through two key designs: PCFA and the shape predictor. The main contribution of UP2You lies in how to rectify unconstrained photos into canonical orthogonal views efficiently and effectively.
>
> ---
>
> > **Q3: Dependency on diffusion priors and potential hallucinations: As a diffusion-based approach, it inherits risks of inconsistent outputs in extreme cases (e.g., highly occluded or sparse inputs), and the paper does not deeply address failure modes or ablation on the limits of rectification without further personalization.**
>
> **TL;DR: UP2You shows robust performance on highly occluded inputs. For sparse inputs with incomplete body coverage, diffusion hallucination may lead to reasonable but improper reconstructions.**
>
> For highly occluded inputs, UP2You effectively extracts and aggregates information from available regions to reconstruct complete 3D humans (Figure 12 (NEW)). When body parts are missing across all input images, diffusion hallucination generates reasonable structures but blends textures from visible parts, reducing appearance accuracy (Figure 13 (NEW)). Under optimal conditions with complete body coverage, UP2You achieves the best quality with minimal hallucination effects (Figures 27-28, Appendix G).

---

> > ### Author Response · Authors · 2025-11-21
> > **Official Response by Authors -- Part 2**
> >
> > > **Q4: Evaluation scope and generalizability: Benchmarks are limited to specific datasets (PuzzleIOI, 4D-Dress, in-the-wild captures); lacks broader comparisons to recent diffusion-free methods or real-time systems. Runtime claims (1.5 minutes) may not hold on lower-end hardware, and virtual try-on is demonstrated but not quantitatively evaluated against specialized try-on models.**
> >
> > **Evaluation Scope**: 3D human reconstruction from unconstrained inputs is a novel task with only PuzzleIOI as a dedicated benchmark. We extend evaluation to 4D-Dress and in-the-wild data, demonstrating generalization across synthetic and real scenarios. As reviewer FySx acknowledged, our experimental evaluation is extensive.
> >
> > **Runtime and Hardware:** We do not claim real-time performance. Existing methods (PuzzleAvatar, AvatarBooth) require 4+ hours; UP2You achieves 1.5 minutes—significantly more efficient.
> >
> > **Virtual Try-On:** We present virtual try-on just as a potential downstream application. UP2You is focused on 3D human reconstruction, not trained on any virtual try-on data. Direct quantitative comparison with specialized try-on models would be unfair, as they target different problem scopes and training objectives.
> >
> > ---
> >
> > > **Q5: Could you elaborate on the novelty of PCFA beyond standard attention mechanisms in diffusion models? How does it differ from similar feature selection in prior works like ReferenceNet or multi-view adapters, and what ablations justify its design over simpler top-k strategies without pose correlation?**
> >
> > Standard attention mechanisms aim to **feature fusion**, treating all features equally, while PCFA is specifically designed for **pose-correlated feature selection** to identify the most informative features. The core motivation behind PCFA is that not all the features from inputs should be treated equally.
> >
> > ReferenceNet and MV-Adapter are feature extraction and multi-view generation modules, respectively, not feature selection methods. Simple top-k selection cannot work without correlation maps predicted by PCFA.
> >
> > ---
> >
> > > **Q6: The pipeline relies on MV-Adapter finetuning for rectification and PSHuman for reconstruction—how much of the performance gains stem from these existing tools versus UP2You's contributions? Would the method work with alternative backbones, or is it tightly coupled to diffusion-based generation?**
> >
> > As mentioned in Q1 and Q2, MV-Adapter and PSHuman are single-view methods incapable of handling unconstrained multi-view inputs. UP2You's core contribution is efficiently rectifying unconstrained photos to canonical orthogonal views. Tables 4 and 11 have already proved the effectiveness of the proposed modules via ablation studies.
> >
> > PCFA is generalizable for pose-correlated feature selection in multiple reference tasks. The current base model MV-Adapter can be replaced with MVDream, Zero123++ and even feed-forward models; and the reconstruction method of PSHuman can be replaced with other reconstruction methods like NeuS or 2DGS. Neither is indispensable.
> >
> > ---
> >
> > > **Q7: For the shape predictor, how robust is it to very sparse inputs (e.g., 1-2 photos with heavy occlusions)? The paper mentions consistency across inputs, but could you provide metrics on shape estimation variance or comparisons to single-image HMR methods like BEDLAM or HYBRIK?**
> >
> > **TL;DR: Table 2 compares the shape predictor with SOTA single-view HMR methods and provides prediction results with sparse inputs (from 3 to 10 photos).**
> >
> > Figure 7 (main paper) and Figure 19 (Appendix E.1) present qualitative results on input images with significant occlusions, illustrating the shape predictor's ability to aggregate information from partially visible views to produce robust shape estimates. Table 10 in Appendix F.3 provides additional analysis of the shape predictor's performance under different input groupings, further demonstrating its robustness.
> >
> > Table 2 includes comparisons with Semantify: The current SOTA shape prediction model, and PromptHMR: The SOTA single-view HMR method, which has been shown to outperform HybrIK. Lastly, we would like to clarify that BEDLAM is a dataset rather than a method, and therefore cannot be directly compared.

---

> ### Comment · Reviewer_VnCH · 2025-11-26
>
> Thank you. I am working on some general 3D reconstruction, so I still have some concerns.
>
> 1. Data Recitifcation.
>
> It can be easily  addressed by Sfm estimation or human pose matching with body template. I think it is not so hard.  We even can use super-glue to guide the matching.
>
> 2. Speed.
>
> I think 1.5 minutes are not so fast. Many 3DGS works can be even faster like pi3 https://github.com/yyfz/Pi3.  Some body template parameter regression methods also can be even faster to serve as an initial pose.
>
> By the way, the speed may attribute to the off-the-shelf components (MV-Adapter and PSHuman) in this work.
>
> 3. Novelty
>
> I still keep my point. The pipeline is somehow adhoc, considering the method heavily builds on existing components (e.g., MV-Adapter for diffusion-based multi-view generation, PSHuman for final mesh reconstruction).

---

> ### Author Response · Authors · 2025-11-27
> **Response to Reviewer VnCH (Continued)**
>
> > **1. Data Recitifcation: It can be easily addressed by Sfm estimation or human pose matching with body template. I think it is not so hard. We even can use super-glue to guide the matching.**
>
> Static scenes do not MOVE or DEFORM, but humans do.
>
> We agree that SfM works well for unconstrained photos of **static scenes**, as shown in Photo Tourism (2006) \[1\]. However, it struggles with clothed and articulated humans due to **dynamic motions and non-rigid clothing deformations**.
>
> In this case, feature descriptors (SIFT, ORB, SuperGlue, etc) are not robust against dynamic wrinkles, the background could be changing, and body parameter regressors are not reliable due to depth ambiguity, making accurate camera or body pose estimation from such unconstrained portraits nearly impossible.
>
> That's why PuzzleAvatar (SIGGRAPH Asia 2024) refers to this as a NEW challenging task, "Album2Human", presents the first benchmark, PuzzleIOI, and throws away the SfM techniques completely in the first place, and strives for camera-free solutions instead. **No off-the-shelf camera estimators are reliable enough for this unconstrained setting**.
>
> If the reviewer knows of any previous work that can "easily" reconstruct **clothed humans from unconstrained photo collections** (w/ various poses, dynamic clothing, and partial visibility) with less than 2 cm errors, within minutes, and produce textured meshes of comparable quality to ours, please mention one. We would appreciate it if the reviewer could highlight any existing methods that can achieve this.
>
> ---
>
> > **2. Speed. I think 1.5 minutes are not so fast. Many 3DGS works can be even faster like pi3 [https://github.com/yyfz/Pi3](https://github.com/yyfz/Pi3). Some body template parameter regression methods also can be even faster to serve as an initial pose.**
>
> Pi3 is point-based, like DUSt3R, not 3DGS-based.
>
> **When discussing speed, we must first clarify the scope.** While some real-time reconstruction methods exist for static scenes and naked bodies from monocular videos or posed photos, no minute-level solution currently exists for reconstructing clothed humans from unconstrained photo collections. The most relevant works, PuzzleAvatar (SIGGRAPH Asia 2024) and the recently published PFAvatar (AAAI 2026) [2], both take hours. **UP2You is the first tuning-free solution for this task, achieving 160-200x speedup over prior art.** And the efficiency of UP2You has been recognized by other reviewers:
>
> - Reviewer zhP9："Additionally, the reconstruction quality and efficiency are demonstrated to be much better than existing SOTAs."
> - Reviewer 8gj9: "Processes unconstrained photos quickly (1.5 minutes total) with nearly constant memory usage regardless of input count."
>
> If the reviewer insists that "1.5 minutes is not fast in contrast with real-time speed," we are happy to rename the title to "**Faster** Reconstruction of Yourself from Unconstrained Photo Collections."
>
> ---
>
> > **3\. Novelty: The pipeline is somehow adhoc, considering the method heavily builds on existing components (e.g., MV-Adapter for diffusion-based multi-view generation, PSHuman for final mesh reconstruction).**
>
> PSHuman combines MVDiffusion and continuous remeshing [3], innovating in body-face cross-scale diffusion for better facial detail reconstruction. We use the standard MVAdaptor with continuous remeshing for a fair comparison, avoiding unnecessary reinvention of the multi-view diffusion model. Additionally, we propose PCFA and a body regressor that scales better with more inputs, eliminating the need for ground truth bodies, unlike PuzzleAvatar.
>
> **AGAIN, neither PSHuman nor MVAdaptor can handle unconstrained photo collections with varying input lengths. Therefore, PCFA is essential for making UP2You functional.**
>
> And our novelties are backed by other reviewers as well:
>
> - Reviewer FySx: "The methodology is well motivated", "The single great insight of this paper is the discovery that correlation maps derived from DINOv2 via regression of its features helps generate multi-view images"
> - Reviewer zhP9: "The authors propose a novel and efficient method for reconstructing 3D humans from unstructured input images", "The proposed correlation map prediction is useful in finding the most informative features, improving the efficiency of the networks."
>
> ---
>
> **References:**
>
> [1] Snavely, Noah, Steven M. Seitz, and Richard Szeliski. "Photo Tourism: exploring photo collections in 3D." SIGGRAPH 2006.
>
> [2] Xi, Dianbing, et al. "PFavatar: Pose-fusion 3d personalized avatar reconstruction from real-world outfit-of-the-day photos." AAAI 2025.
>
> [3] Palfinger, Werner. "Continuous remeshing for inverse rendering." Computer Animation and Virtual Worlds 2022.

---

> ### Comment · Reviewer_VnCH · 2025-11-28
>
> Thanks for the clarification. It solves most concerns.
>
> One question remains. "the speed may attribute to the off-the-shelf components (MV-Adapter and PSHuman) in this work." Could you add or refer to the time cost of each components? and compare with the similar-function sub-component in previous works?
>
> If so, would you like to modify the first contribution on efficiency to avoid the potential overclaim, considering the efficiency is mostly credited to previous works?

---

> > ### Author Response · Authors · 2025-11-28
> > **Response to Reviewer VnCH**
> >
> > Additionally, following your suggestion, we’ve added a table comparing the module latency of UP2You, MV-Adapter, and PSHuman below:
> >
> > | Method | Shape Prediction | Multi-View Image Generation | Multi-View Normal Generation | Reconstruction (Continuous Remeshing) |
> > | ----- | ----- | ----- | ----- | ----- |
> > | UP2You | 0.02s | 15s | 15s | \~1min |
> > | PSHuman | N/A | 25s | Together with Image | \~1.2min |
> > | MV-Adapter | N/A | 12s | N/A | N/A |
> >
> > Notes: PSHuman jointly generates images and normals, leading to 25s for one case. Compared with UP2You, PSHuman needs to further optimize SMPL-X parameters, which will slightly increase the reconstruction time to 1.2 min. MV-Adapter only takes one image as the condition to generate multi-view images, taking the least generation time of 12s.
> >
> > ---
> >
> > Thank you for your input. We will revise the first contribution to better credit MV-Adaptor and continuous remeshing:
> >
> > As shown in Fig. 2, unlike previous tuning-based works (fine-tuning diffusion model and running SDS optimization for over 4 hours), UP2You is a tuning-free solution that directly generates "clean" multi-views from "dirty" inputs via feed-forward multi-view diffusion (MV-Adaptor, 30 secs) and reconstructs the textured mesh with continuous remeshing (1 min). With such a new “data rectifier” paradigm, it can handle one or dozens of photos with a nearly constant memory footprint. The entire pipeline completes in 1.5 minutes.
> >
> > We will include this efficiency analysis in our revised paper. If this addresses your concerns, please give us a thumbs-up and update the ratings accordingly.
> >
> > We're always on-call for further questions.

---

### Official Review · Reviewer_8gj9 · 2025-10-31

**Soundness:** 3
**Presentation:** 3
**Contribution:** 3
**Rating:** 8
**Confidence:** 4

**Summary:**

UP2You is a tuning-free method for reconstructing high-fidelity 3D clothed portraits from unconstrained 2D photos. It uses a "data rectifier" paradigm (via modules like PCFA and a perceiver-based shape predictor) to convert raw, unstructured inputs into orthogonal multi-view images and normal maps, enabling fast (1.5 minutes per person) and high-quality reconstruction. It outperforms baselines on geometric accuracy and texture fidelity across datasets like PuzzleIOI and 4D-Dress.

**Strengths:**

1. Processes unconstrained photos quickly (1.5 minutes total) with nearly constant memory usage regardless of input count.
2. Surpasses SOTA methods in key metrics (e.g., 15% lower Chamfer distance on PuzzleIOI, 21% higher PSNR on 4D-Dress) via selective feature aggregation.
3. Supports arbitrary pose control and training-free 3D virtual try-on without relying on pre-captured body templates.

**Weaknesses:**

1. In 3D clothed portrait reconstruction, how does the rectifier ensure data quality when converting unconstrained photos into orthogonal views? What mechanisms maintain multi-view texture consistency and preserve fine details like clothing patterns and facial features, avoiding distortion or misalignment?

2. For unconstrained inputs with unknown or noisy camera parameters, how does the rectifier align orthogonal views with accurate intrinsic and extrinsic settings? Does it use camera calibration or pose/shape priors (e.g., SMPL-X normal maps)? Is the alignment achieved via iterative optimization or direct mapping?

3. How does this rectification improve the geometric accuracy of subsequent 3D reconstruction, such as reducing mesh carving or texture baking errors?

4. The paper examines reference image count and feature aggregation (e.g., averaging vs. PCFA) but lacks ablation on key PCFA hyperparameters—like γ (feature retention ratio) or transformer block number. How do these affect reconstruction accuracy (e.g., smaller γ causing detail loss or larger γ adding noise), and why were the final values chosen?

**Questions:**

Please refer to the weakness part.

**Details Of Ethics Concerns:**

N/A.

---

> ### Author Response · Authors · 2025-11-21
> **Official Response by Authors**
>
> We would like to sincerely thank the reviewer for the detailed and constructive comments on our work. We take every comment seriously and hope our response can address the reviewer’s concerns. If there are any remaining questions, we are more than happy to address them.
>
> ---
>
> > **Q1: In 3D clothed portrait reconstruction, how does the rectifier ensure data quality when converting unconstrained photos into orthogonal views? What mechanisms maintain multi-view texture consistency and preserve fine details like clothing patterns and facial features, avoiding distortion or misalignment?**
>
> **TL;DR: We ensure high-quality rectification through supervised training, view-consistent synthetic rendering data, multi-view attention mechanism, strong ReferenceNet encoder to capture human identity, and PCFA module for multi-input fusion.**
>
> UP2You is trained on synthetic human scans with SMPL-X ground-truth (Appendix D). During training, the rectifier is supervised by rendering orthogonal view images, while the inputs are randomly rendered images from different human and camera poses of the same identity. This training objective encourages the rectifier to learn to transform unconstrained input photos to accurate pose-conditioned orthogonal views.
>
> The row-wise self-attention in MV-Adapter maintains cross-view consistency for orthogonal views. The ReferenceNet encoder extracts fine-grained texture details (e.g., clothing patterns, facial features), and the high output resolution (768×768) also reduces distortion. The PCFA module further provides proper reference features for each orthogonal view, reducing misalignment.
>
> ---
>
> > **Q2: For unconstrained inputs with unknown or noisy camera parameters, how does the rectifier align orthogonal views with accurate intrinsic and extrinsic settings? Does it use camera calibration or pose/shape priors (e.g., SMPL-X normal maps)? Is the alignment achieved via iterative optimization or direct mapping?**
>
> **TL;DR: UP2You does not require estimated or optimized camera parameters at all. Orthogonal cameras are pre-defined as conditions for multi-view generation. No camera alignment or any form of optimization is needed.**
>
> During inference, UP2You estimates shape parameters, then renders the SMPL-X mesh in predefined canonical A-Pose using pre-defined orthogonal cameras (0° elevation, azimuths in {0°, 45°, 90°, 135°, 180°, 270°}). The rendered SMPL-X normals serve as view conditions for rectifying unconstrained inputs to target orthogonal views.
>
> ---
>
> > **Q3: How does this rectification improve the geometric accuracy of subsequent 3D reconstruction, such as reducing mesh carving or texture baking errors?**
>
> **TL;DR: Without rectification, cameras or body pose have to be estimated accurately, which is almost impossible when it comes to unconstrained inputs.**
>
> Rectification improves geometric accuracy by transforming unconstrained inputs into predefined human and camera poses, eliminating the need to predict SMPL-X pose and camera parameters. Since unconstrained inputs exhibit severe occlusion and large pose variations, direct pose and camera estimation are highly challenging and error-prone, leading to poor reconstruction quality. By bypassing this unreliable estimation step, rectification ensures consistent and accurate orthogonal views for robust 3D reconstruction.
>
> ---
>
> > **Q4: The paper examines reference image count and feature aggregation (e.g., averaging vs. PCFA) but lacks ablation on key PCFA hyperparameters—like $\gamma$ (feature retention ratio) or transformer block number. How do these affect reconstruction accuracy (e.g., smaller $\gamma$ causing detail loss or larger $\gamma$ adding noise), and why were the final values chosen?**
>
> **TL;DR: $\gamma= 2.0$ achieves the best trade-off between generation quality and GPU memory efficiency.**
>
> We set the default value of $\\gamma$ to 2.0 to control the number of reference features selected in the top-k selection. To determine the optimal configuration, we evaluate different values of $\\gamma$ below (updated in Table 7 (NEW) of the revised paper). The results demonstrate that $\\gamma=2.0$ achieves the best trade-off between generation quality and GPU memory efficiency.
>
> |  | PSNR$\\uparrow$ | SSIM$\\uparrow$ | LPIPS$\\downarrow$ | GPU$\\downarrow$ |
> | :---- | :---- | :---- | :---- | :---- |
> | $\\gamma=1.0$ | 24.978 | 0.912 | 0.0665 | **18.64** |
> | $\\gamma=2.0$ | **25.848** | **0.920** | **0.0576** | 20.88 |
> | $\\gamma=3.0$ | 25.837 | **0.920** | 0.0569 | 23.12 |

---

> ### Author Response · Authors · 2025-11-27
> **Looking Forward to Your Feedback**
>
> Dear Reviewer,
>
> I hope you're doing well. With **less than a week left** in the discussion period, I want to ensure we've addressed all your concerns. If you have any additional feedback or points for us to consider, please let us know. Your insights are invaluable, and we're keen to resolve any remaining issues to enhance our work.
>
> Thank you for your time and effort in reviewing our paper.

---

### Official Review · Reviewer_zhP9 · 2025-11-01

**Soundness:** 3
**Presentation:** 3
**Contribution:** 3
**Rating:** 8
**Confidence:** 4

**Summary:**

This paper introduces a new method, called UP2You, for reconstructing 3D humans from unstructured inputs with various poses, cameras, and visible body parts. To achieve this goal, the authors separate the procedure into two parts: multi-view orthogonal images generation and 3D mesh reconstruction. The major point lies in the generation of high-quality multi-view images. Specifically, (1) the authors propose a correlation map prediction that allows for selecting the most informative features from input images. Furthermore, this technique helps maintain the computational efficiency while having more input images. (2) MVDiffusion model is applied to generate the multi-view images based on the selected feature, pose, and normal maps. Extensive experiments have been proposed to demonstrate the performance of UP2You, presenting much better performance than existing methods.

**Strengths:**

+ The authors propose a novel and efficient method for reconstructing 3D humans from unstructured input images. Additionally, the reconstruction quality and efficiency are demonstrated to be much better than existing SOTAs.

+ The proposed correlation map prediction is useful in finding the most informative features, improving the efficiency of the networks.

+ The paper is well-writen and easy to follow

**Weaknesses:**

The reviewer believe the method is well-designed and may have some concerns regarding the experiments:

- The method lacks the qualitative and quantitative comparisons with standard human reconstruction, such as PIFu, ICON, ECON. This would help demonstrate the reason why we need this type of method with unstructured inputs.

- The paper would benefit from more results with loose clothing, complex poses as in ECON to demonstrate the performance.

- The paper may also benefit from the evaluation of the pose variations and occlusion ratio presented in the input? For example, (1) can UP2You handle inputs with largely different poses? (2) what if some parts of the human body are not presented in the input images? (3) what's the maximal occlusion ratio in the used test set? One example might be: the input contains an image which captures only the foot part, while the other images don't contain the foot.

**Questions:**

Besides the weaknesses listed above, the reviewer has one more question: is the current version of UP2You capable of animating the reconstructed human mesh, considering the input contains various poses?

---

> ### Author Response · Authors · 2025-11-21
> **Official Response by Authors -- Part 1**
>
> We would like to sincerely thank the reviewer for the detailed and constructive comments on our work. We take every comment seriously and hope our response can address the reviewer’s concerns. If there are any remaining questions, we are more than happy to address them.
>
> ---
>
> > **Q1: The method lacks the qualitative and quantitative comparisons with standard human reconstruction, such as PIFu, ICON, ECON. This would help demonstrate the reason why we need this type of method with unstructured inputs.**
>
> **TL;DR: We added comparisons with standard single-view methods (ICON, ECON, PIFuHD, Human3Diff, PSHuman). UP2You with unconstrained photos outperforms all single front-view based methods, especially in back-view generation.**
>
> We initially compared UP2You with PSHuman (SOTA method for single-view human reconstruction) based on single-view inputs for fair comparison, demonstrating generalization but not highlighting the advantages of unconstrained inputs.
>
> Following the suggestion, we now compare against standard human reconstruction methods using unconstrained photos as UP2You input. Our method achieves superior rendering quality and 3D accuracy, particularly for back views, validating the value of unconstrained inputs (more accessible, more informative). Results are presented in Table 8 (NEW) and below:
>
> | Method | PSNR Front$\uparrow$ | PSNR Back$\uparrow$ | SSIM Front$\uparrow$ | SSIM Back$\uparrow$ | LPIPS Front$\downarrow$ | LPIPS Back$\downarrow$ | Chamfer$\downarrow$ | P2S$\downarrow$ | Normal$\downarrow$ |
> | :---- | ----: | ----: | ----: | ----: | ----: | ----: | ----: | ----: | ----: |
> | Ours\* | 25.257 | **25.488** | **0.906** | **0.909** | **0.0724** | **0.0733** | **1.140** | **1.119** | **0.0122** |
> | PSHuman | **25.384** | 23.382 | 0.898 | 0.885 | 0.0934 | 0.1121 | 2.756 | 2.926 | 0.0189 |
> | Human3Diff | 23.335 | 20.720 | 0.883 | 0.872 | 0.1118 | 0.1248 | 4.275 | 4.322 | 0.0227 |
> | ICON | \- | \- | \- | \- | \- | \- | 4.352 | 4.331 | 0.0188 |
> | ECON | \- | \- | \- | \- | \- | \- | 3.780 | 3.642 | 0.0178 |
> | PIFuHD | \- | \- | \- | \- | \- | \- | 2.776 | 2.603 | 0.0154 |
>
> ---
>
> > **Q2: The paper would benefit from more results with loose clothing, complex poses as in ECON to demonstrate the performance.**
>
> **See Figures 23 and 24 (New) in Appendix F.2.**
>
> Figure 23 in Appendix F.2 provides generated results with diverse poses and clothing styles. To further demonstrate the performance of UP2You on subjects with loose clothing and complex target poses, we have added more results in Figure 24 of Appendix F.2.
>
> ---
>
> > **Q3: The paper may also benefit from the evaluation of the pose variations and occlusion ratio presented in the input? For example, (1) can UP2You handle inputs with largely different poses? (2) what if some parts of the human body are not presented in the input images? (3) what's the maximal occlusion ratio in the used test set? One example might be: the input contains an image which captures only the foot part, while the other images don't contain the foot.**
>
> (1) **Inputs with Largely Different Poses:** **Figures 27 and 28** in Appendix G demonstrate that UP2You handles diverse human and camera poses.
>
> (2) **Missing Body Parts in Inputs:** **Figure 13 (NEW)** examines scenarios where body parts are not fully visible across all images (e.g., the foot region). Due to diffusion hallucination, the generated results exhibit a somewhat reasonable structure; however, the texture is blended from other visible parts. Therefore, inputs with complete body part coverage are more suitable for UP2You to achieve optimal results.
>
> (3) **High Occlusion Ratios in Inputs:** **Figure 12 (NEW)** presents an example where one input image captures only the foot region, while other images lack this body part, demonstrating our method's capability to handle inputs with high occlusion ratios. More input samples of the test set are shown in **Figures 27 and 28** of Appendix G.

---

> > ### Author Response · Authors · 2025-11-21
> > **Official Response by Authors -- Part 2**
> >
> > > **Q4: Besides the weaknesses listed above, the reviewer has one more question: is the current version of UP2You capable of animating the reconstructed human mesh, considering the input contains various poses?**
> >
> > **TL;DR: The reconstructed mesh is easily animatable since UP2You adopts A-Pose as the default target pose and provides aligned SMPL-X model.**
> >
> > As we take A-Pose as the default target pose, it is easy to animate an A-Pose human mesh with third-party animation tools like Mixamo \[1\]. Figure 7 of PuzzleAvatar \[2\] has provided a visual example, and we also add an animation case in **Figure 21 (NEW)** of Appendix E.3. Meanwhile, UP2You also provides the aligned SMPL-X of the reconstructed human; methods based on skin weight transfer \[3\] can also be applied for animation.
> >
> > \[1\] Blackman, Sue. "Rigging with mixamo." Unity for Absolute Beginners. Berkeley, CA: Apress, 2014. 565-573\.
> >
> > \[2\] Xiu, Yuliang, et al. Puzzleavatar: Assembling 3d avatars from personal albums. TOG 2024\.
> >
> > \[3\] Abdrashitov, Rinat, et al. Robust skin weights transfer via weight inpainting. SIGGRAPH Technical Communications 2023\.

---

> > > ### Comment · Reviewer_zhP9 · 2025-11-24
> > >
> > > Thanks for the detailed responses from the authors, which have addressed most of my initial concerns. Here are some further comments:
> > >
> > > 1. For Figures 23 and 24, it would be better to provide results from different viewpoints, instead of only the frontal view.
> > >
> > > 2. For Figure 13, what if other critical body parts—such as the legs or portions of the upper body—are missing? These cannot be trivially substituted with SMPL?
> > >
> > > 3. From the visualizations in Figure 21, the facial region is not very realistic, and more like a texture mapping with few geometric details. The animation via Mixamo is not what I wanted to mean, as it typically leads to bad results for clothes, hair, accessories, etc. I was wondering whether the reconstruction could instead be animated directly using the UP2You pre-trained network.

---

> > > > ### Author Response · Authors · 2025-11-25
> > > > **Response to Reviewer zhP9 (Continued)**
> > > >
> > > > > **For Figures 23 and 24, it would be better to provide results from different viewpoints, instead of only the frontal view.**
> > > >
> > > > We have updated Figures 23 and 24 in the revised manuscript to include results from multiple viewpoints (6 orthogonal views), instead of the front-back view.
> > > >
> > > > ---
> > > >
> > > > > **For Figure 13, what if other critical body parts—such as the legs or portions of the upper body—are missing? These cannot be trivially substituted with SMPL?**
> > > >
> > > > We have added extensive results in Figure 29 (NEW) of Appendix G, examining UP2You's performance with different missing body parts, including the face, lower body, and upper body. These results show that UP2You is robust to diverse and even extreme input scenarios, successfully handling cases where critical body parts are not visible in the input images.
> > > >
> > > > ---
> > > >
> > > > > **From the visualizations in Figure 21, the facial region is not very realistic, and more like a texture mapping with few geometric details. The animation via Mixamo is not what I wanted to mean, as it typically leads to bad results for clothes, hair, accessories, etc. I was wondering whether the reconstruction could instead be animated directly using the UP2You pre-trained network.**
> > > >
> > > > **Rendering Quality:** UP2You generates textured meshes with vertex colors, which are not well supported by Mixamo's renderer.
> > > >
> > > > **Animation Using UP2You Network:** Following your suggestion, we have updated Figure 21 with animation results generated directly using the UP2You pre-trained network. Since UP2You can rectify unconstrained inputs into different target pose configurations, we leverage this capability to generate animated sequences by reconstructing the mesh in various poses. The results demonstrate that UP2You maintains excellent identity consistency across different target poses. However, as this approach independently reconstructs a textured mesh for each frame, temporal consistency of both the rendered images and mesh topology cannot be guaranteed. This is an interesting direction for future work.

---

### Official Review · Reviewer_FySx · 2025-11-02

**Soundness:** 3
**Presentation:** 3
**Contribution:** 2
**Rating:** 4
**Confidence:** 3

**Summary:**

This paper proposes a method for 3D reconstruction from a human mesh from relatively unconstrained photographs with the only requirement being that the person must be wearing the same clothes. The overall pipeline consists of three stages:

1) First, from unconstrained photographs, six full body views are generated (with the appropriate inpainting)
2) Second, using the aforementioned images and coarse SMPL normals, detailed normals are predicted
3) Finally, features from the first step are used to refine the global shape using a perceiver like architecture.

Qualitative and quantitatively the proposed method outperforms prior work and is relatively fast.

**Strengths:**

1) The paper is well written and easy to follow
2) The methodology is well motivated
3) The single great insight of this paper is the discovery that correlation maps derived from DINOv2 via regression of its features helps generate multi-view images
4) Strong quantitative and qualitative results

**Weaknesses:**

One weakness of this paper is the lack of proper ablation regarding two of it’s non-trivial decisions

1) Why DINOv2 features? A plethora of semantic segmentation networks exists (including DINOv1) and at least some of them must be ablation to demonstrate the efficacy of constructing features maps via DINOv2

2) The justification for using the Perciever model for shape refinement if relatively ad-hoc, it would be great if the authors could also share with more baselines such as a transferformer or a few MLPs

**Questions:**

See weaknesses

---

> ### Author Response · Authors · 2025-11-21
> **Official Response by Authors**
>
> We sincerely thank the reviewer for acknowledging our contributions and for providing valuable questions and suggestions.  We take every comment seriously and hope our response can address the reviewer’s concerns. If there are any remaining questions, we are more than happy to address them.
>
>
> >**Q1: Why DINOv2 features?**
>
> **TL;DR: We chose DINOv2 for its view-consistency property as noted in FoundPose [1], and proven effectiveness in capturing pose-related features [2,3,4].**
>
> In the PCFA module, we want to establish correspondences between unconstrained input photos and target canonical views to select the most informative features. DINOv2 has been proven effective at capturing correspondences between 3D objects and their renderings [1], making it ideal for PCFA.
>
> Since target orthogonal view images are unavailable at inference, we use SMPL-X poses as proxies. Although this shifts from 2D-to-3D to 2D-to-Pose correspondence, DINOv2 remains highly effective for extracting pose-related features. Recent human mesh recovery methods [2-4] validate this, demonstrating DINOv2's ability to extract rich pose-relevant information without labeling.
>
> [1] Örnek, Evin Pınar, et al. Foundpose: Unseen object pose estimation with foundation features. ECCV 2024.
>
> [2] Su, Chi, et al. Sat-hmr: Real-time multi-person 3D mesh estimation via scale-adaptive tokens. CVPR 2025.
>
> [3] Baradel, Fabien, et al. Multi-hmr: Multi-person whole-body human mesh recovery in a single shot. ECCV 2024.
>
> [4] Wang, Yufu, et al. Prompthmr: Promptable human mesh recovery. CVPR 2025.
>
> ---
>
> > **Q2: A plethora of semantic segmentation networks exists (including DINOv1), and at least some of them should be ablated to demonstrate the efficacy of constructing feature maps via DINOv2.**
>
> **TL;DR: Ablation study shows DINOv2 outperforms DINOv1 and CLIP in modeling pose-related correlation maps.**
>
> We have included an ablation study comparing DINOv1, CLIP, and DINOv2 as image encoders. The results below (new Table 6 in revised paper) demonstrate DINOv2's superior performance for multi-view image generation:
>
> |  | PSNR$\\uparrow$ | SSIM$\\uparrow$ | LPIPS$\\downarrow$ |
> | :---- | :---- | :---- | :---- |
> | **DINOv2** | **25.848** | **0.920** | **0.0576** |
> | CLIP | 23.876 | 0.904 | 0.0767 |
> | DINOv1 | 24.170 | 0.907 | 0.0745 |
>
> ---
>
> > **Q3: The justification for using the Perceiver model for shape refinement is relatively ad-hoc; it would be great if the authors could also compare with more baselines such as a Transformer or a few MLPs.**
>
> **TL;DR: Perceiver is designed to handle variable-length input, which is MLP short for.**
>
> The Perceiver is a Transformer-based architecture that employs latent units to summarize variable-length data, which is ideal for our shape refinement module. In UP2You, learnable shape tokens serve as latent units, while DINOv2 features from multiple reference images are keys/values in cross-attention. This design effectively aggregates variable-length features into compact shape tokens for parameter prediction.
>
> MLPs, by contrast, struggle with variable-length inputs since **averaging operations inherently lose information**.
>
> We implemented an MLP-based baseline for shape refinement, where the DINOv2 features from multiple reference images were averaged to predict the shape parameters. Results are shown below and added in Table 2 of the revised paper. The results show MLP-based structure performs worse than the Perceiver-based design, and the performance degrades as the number of reference images increases, due to the information loss from averaging.
>
> | Num of References |  MLP | Ours |
> |------------|------|------|
> | 3 |  8.819 | **7.967** |
> | 6 | 8.046 | **7.427** |
> | 9 | 8.275 | **7.403** |
> | 12 | 8.336 | **7.399** |
>
> **V2V$\downarrow$ Comparisons of Shape Prediction Results.**

---

> ### Author Response · Authors · 2025-11-27
> **Looking Forward to Your Feedback**
>
> Dear Reviewer,
>
> I hope you're doing well. With **less than a week left** in the discussion period, I want to ensure we've addressed all your concerns. If you have any additional feedback or points for us to consider, please let us know. Your insights are invaluable, and we're keen to resolve any remaining issues to enhance our work.
>
> Thank you for your time and effort in reviewing our paper.

---

> > ### Comment · Reviewer_FySx · 2025-11-28
> >
> > Thanks for the new results and the response! After looking at other reviews and taking into consideration the new results I have increased my rating

---

> ### Author Response · Authors · 2025-11-28
>
> We’re really happy our response addresses your concerns!
>
> It seems there might be a bug on the OpenReview platform since the updated rating isn’t showing. Could you please confirm your final rating, explicitly, in your last comment? It would really help the ACs and PC with their decision. Thanks so much!

---

### Author Response · Authors · 2025-11-21
**General Response to All Reviewers**

Dear Reviewers:

We would like to thank you for your constructive comments, which are extremely helpful for improving our paper. We have uploaded a revised draft of our paper.

The major updates are as follows:
+ Additional results of the shape predictor with MLP structure in Table 2.
+ Visual results of subjects with loose clothing and complex target poses in Figure 24 of Appendix F.2.
+ Added ablation study about different image encoder choices of PCFA, results are shown in Table 6.
+ Added ablation study about the number of selected features $\gamma$, results are shown in Table 7.
+ Comparisons between UP2You with unconstrained photos and standard 3D human reconstruction methods with a single front view image, shown in Table 8.
+ Results with high occlusion in Figure 12.
+ Results with missing part in Figure 13 and Figure 29.
+ Animation analysis and visual results in Figure 21.
+ Change Figure 23 to provide results with more viewpoints.

Please do feel free to let us know if you have any further questions.

---

### Author Response · Authors · 2025-12-01
**General Response (3/3)**

## Part III: Key Highlights from New Experiments
To address concerns regarding inputs, ablations, and robustness, we conduct comprehensive new experiments. We summarize the three most key findings below:

### **1. Inputs: Unconstrained Photos vs. Single Front-view Image (R2)**
*UP2You with unconstrained photos vs. standard methods with single front-view image*
We clarify the value of unconstrained inputs (more accessible, more informative), especially in back-view results.
| Method | PSNR Front$\uparrow$ | PSNR Back$\uparrow$ | SSIM Front$\uparrow$ | SSIM Back$\uparrow$ | LPIPS Front$\downarrow$ | LPIPS Back$\downarrow$ | Chamfer$\downarrow$ | P2S$\downarrow$ | Normal$\downarrow$ |
| :----: | :----: | :----: | :----: | :----: | :----: | :----: | :----: | :----: | :----: |
| Ours\* | 25.257 | **25.488** | **0.906** | **0.909** | **0.0724** | **0.0733** | **1.140** | **1.119** | **0.0122** |
| PSHuman | **25.384** | 23.382 | 0.898 | 0.885 | 0.0934 | 0.1121 | 2.756 | 2.926 | 0.0189 |
| Human3Diff | 23.335 | 20.720 | 0.883 | 0.872 | 0.1118 | 0.1248 | 4.275 | 4.322 | 0.0227 |
| ICON | \- | \- | \- | \- | \- | \- | 4.352 | 4.331 | 0.0188 |
| ECON | \- | \- | \- | \- | \- | \- | 3.780 | 3.642 | 0.0178 |
| PIFuHD | \- | \- | \- | \- | \- | \- | 2.776 | 2.603 | 0.0154 |


---


### **2. Ablations: Image Encoder & Shape Predictor & Feature Selection (R1, R3)**
*Reviews suggested analysis on different image encoders of PCFA (R1), different backbones of shape predictor (R1), and different feature selection ratios (R3).*

**A. Different Image Encoders in PCFA:** DINOv1 vs. CLIP vs. DINOv2.
|  | PSNR$\uparrow$ | SSIM$\uparrow$ | LPIPS$\downarrow$ |
| :---- | :---- | :---- | :---- |
| **DINOv2** | **25.848** | **0.920** | **0.0576** |
| CLIP | 23.876 | 0.904 | 0.0767 |
| DINOv1 | 24.170 | 0.907 | 0.0745 |

**B. Different Backbones of Shape Predictor:** MLP vs. Perceiver Transformer. Perceiver is designed to handle variable-length input, which is MLP short for.

**v2v$\downarrow$ comparisons of shape prediction results:**
| Num of References |  MLP | Ours (Perceiver) |
|:------------:|:------:|:------:|
| 3 |  8.819 | **7.967** |
| 6 | 8.046 | **7.427** |
| 9 | 8.275 | **7.403** |
| 12 | 8.336 | **7.399** |

**C. Different Feature Selection Ratio $\gamma$:** $\gamma=2.0$ achieves the best trade-off between generation quality and GPU memory efficiency.
|  | PSNR$\uparrow$ | SSIM$\uparrow$ | LPIPS$\downarrow$ | GPU$\downarrow$ |
| :---- | :---- | :---- | :---- | :---- |
| $\gamma=1.0$ | 24.978 | 0.912 | 0.0665 | **18.64** |
| $\gamma=2.0$ | **25.848** | **0.920** | **0.0576** | 20.88 |
| $\gamma=3.0$ | 25.837 | **0.920** | 0.0569 | 23.12 |

---

### **3. Robustness Analysis (R2)**
We add new qualitative results on challenging scenarios:
|Challenge|Visualization|Conclusion|
|:---|:---:|:---|
|**Loose Clothing & Hard Pose**| `Figure 24`|Robust to challenging clothing and poses|
|**Missing Body Parts**|`Figure 13`, `Figure 29`|Robust to missing body parts: Reasonable geometry, and texture may copied from visible part.|
|**High Occlusion Ratios in Inputs**|`Figure 12`|Robust to high occlusion in inputs|

---

### Author Response · Authors · 2025-12-01
**General Response (2/3)**

## Part II: Summary of Rebuttal

We make a summary of the rebuttal results so far in the table below, and we would also like to kindly remind you that **R1(FySx)** **promised to raise scores**, and **R4(VnCH)** admitted that **"It solves most concerns"** before being reverted.

|Reviewer|Orignal Rating/Confidence|Concerns|Rebuttal|Final Attitude|
|:---:|:---:|:--|:--|:---|
|**R1(FySx)**|4/3| **(1)** Add ablation study of image encoder selection in PCFA. **(2)** Add ablation study of backbone selection in shape predictor.|**(1)** Added ablation study of image encoder selection in `Table 6`. **(2)** Added ablation study of shape predictor backbone in `Table 2`.|**Promised to raise scores**|
|**R2(zhP9)**|8/4|**(1)** Add more comparisons with standard human reconstruction methods. **(2)** Add more results with loose clothing, complex poses. **(3)** Eval more results with pose variations and high occlusion. **(4)** Add animation results.|**(1)** Added more comparisons in `Table 8`. **(2)** Added results with loose clothing and complex poses in `Figure 24`. **(3)** Added evaluation results with high occlusion in `Figure 12`, `Figure 13`, and `Figure 29`. **(4)** Added animation results in `Figure 21`.|**Addressed most of initial concerns**|
|**R3(8gj9)**|8/4|**(1)** Some questions about the designs and fuctions of UP2You. **(2)** Add ablation study of different $\gamma$ (feature selection ratio).|**(1)** Clarified the functions and procedures of UP2You. **(2)** Added ablation study of different $\gamma$ in `Table 7`.|-|
|**R4(VnCH)**|4/3|**(1)** Questions about novelty, designs, experiments, and efficiency of UP2You. **(2)** Add speed analysis to clarify the efficiency contribution.|**(1)** Responded to all relevant questions. **(2)** Added speed analysis of different components, and emphasized our contribution in the paradigm.|**Solved most concerns**|

For more details, please refer to the point-by-point responses in each reviewer’s thread.

---

### Author Response · Authors · 2025-12-01
**General Response (1/3)**

We thank **R1(FySx)**, **R2(zhP9)**, **R3(8gj9)**, **R4(VnCH)** for their constructive feedback and discussions. We are encouraged by their consensus on the contribution of our work. UP2You code will be released for future research.

Common recognition of our strengths includes:
1. Novelty & Contribution:
Recognized as a **"well motivated methodology with great insight"** (R1) and **"a novel and efficient method"** (R2) for reconstructing 3D humans from unstructured input images. R3 and R4 also recognize the **"nearly constant memory usage"** regardless of input count.

2. Strong Performance:
**Strong quantitative and qualitative results** (R1) demonstrate UP2You is **much better than existing SOTAs** (R2, R3).

3. Efficiency:
**"Efficient Method"** (R2), **"processes unconstrained photos quickly"** (R3), and **"enables fast reconstruction"** (R4).


We have conducted extensive additional experiments (**4 Tables and 6 Figures**) and analysis to address each concern of reviewers, and revised the manuscript accordingly. All changes in the updated `PDF` are highlighted in **red**.

---

## Part I: Summary of Revisions
1. **Component Analysis (R1, R3)**
    + **Different image encoders** in PCFA (DINOv1 vs. CLIP vs. DINOv2) (R1-Q1; `Table 6`).
    + **Different backbones** of shape predictor (MLP vs. Perceiver Transformer) (R1-Q2; `Table 2`).
    + **Different feature selection ratios** $\gamma$ in PCFA (R3-Q1; `Table 7`).
2. **Additional Comparisons with Standard Methods (R3)**
    + **Quantitative comparisons** with PIFuHD, ICON, ECON, PSHuman, Human3Diff (R2-Q1; `Table 8`).
3. **Robustness Analysis (R2)**
    + **Loose clothing and complex poses** of input identity (R2-Q2; `Figure 24`).
    + **Results with More Viewpoints** to show results under diverse target pose (R2-Q2, R2-Continued-Q1; `Figure 23`).
    + **High occlusion** in unconstrained photos (R2-Q3; `Figure 12`, `Figure 13`, `Figure 29`).

4. **Animation Results (R2)**
    + **Animation with UP2You** to show animation capability (R2-Q4; `Figure 21`).

---

### Meta-Review · Area_Chair_vFGA · 2026-01-04

**Summary:**

The reviewers generally agreed that this paper tackles an important and underexplored problem: reconstructing high-fidelity 3D clothed humans from unconstrained photo collections, with strong quantitative and qualitative performance, and a favorable efficiency-quality trade-off compared to prior methods.

The main concerns informing the initial mixed scores were:
- missing ablations for key design choices (image encoder in PCFA, backbone of the shape predictor);
- insufficient comparisons with standard human reconstruction baselines (e.g., ICON, ECON, PIFuHD);
- limited robustness analysis under challenging conditions such as loose clothing, extreme poses, high occlusion, and missing body parts;
- questions about novelty and efficiency attribution, given the reliance on existing components (MV-Adapter, PSHuman).

The rebuttal and discussion phase substantially strengthened the paper through new experiments, analyses, and clarifications, addressing most of these concerns. Considering the paper's effectiveness and efficiency, AC recommends acceptance.

**Reviewer Concerns:**

## Addressed concerns

**Design justification (FySx, 8gj9):**
The authors added comprehensive ablations on PCFA image encoder choices (DINOv2 vs. DINOv1 vs. CLIP), shape predictor backbones (Perceiver vs. MLP), and the feature selection ratio.

**Evaluation and novelty (zhP9):**
New quantitative comparisons against standard human reconstruction methods (ICON, ECON, PIFuHD, Human3Diff, PSHuman) were added.

**Robustness analysis (zhP9, 8gj9):**
Additional qualitative and quantitative results were provided for loose clothing, complex poses, high occlusion, and missing body parts, as well as multi-view visualizations from different viewpoints. These additions substantially improved the robustness evaluation.

**Efficiency attribution (VnCH):**
The authors added a component-wise runtime breakdown and revised the wording of their efficiency claims to more clearly credit reused components, while clarifying the specific contribution of the rectification paradigm and PCFA in enabling efficient handling of unconstrained, variable-length inputs.

## Outstanding concerns

**Extreme failure modes and animation consistency (zhP9, VnCH):**
While the authors explicitly analyzed and acknowledged hallucination effects when critical body parts are missing across all inputs, and demonstrated pose-conditioned animation via repeated reconstruction, true temporal and topological consistency across animation frames remains an open limitation. This is appropriately acknowledged as future work and does not negate the main contribution.

**Reviewer Scores:**

Reviewer FySx:
Initial score: 4 | Estimated final score: 6.
After adding ablations and discussion, the reviewer explicitly stated the increase in rating.

Reviewer zhP9:
Initial score: 8 | Estimated final score: 8.
Most concerns were addressed with additional comparisons, robustness tests, and clarification of animation capabilities.

Reviewer 8gj9:
Initial score: 8 | Estimated final score: 8.
Requested clarifications and hyperparameter ablations were addressed.

Reviewer VnCH:
Initial score: 4 | Estimated final score: 6.
After extensive discussion, added analyses, and efficiency breakdown, the reviewer stated that “it solves most concerns.”

---

### Decision · Program_Chairs · 2026-01-26

Accept (Poster)